# Determining the distance patterns in the movements of future doctors in UK between 2002 and 2015: a retrospective cohort study

Lucy Hitchings,[1] Ben Fleet [ID],[2] Daniel Thomas Smith [ID],[3] Jonathan M Read [ID],[4] Colin Melville,[5,6] Luigi Sedda [ID][7]

LH and BF are joint first authors.

For numbered affiliations see end of article.

**Correspondence to**
Dr Luigi Sedda;
l.sedda@lancaster.ac.uk

## ABSTRACT

**Objective** To determine and identify distance patterns in the movements of medical students and junior doctors between their training locations.

**Design** A retrospective cohort study of UK medical students from 2002 to 2015 (UKMED data).

**Setting** All UK medical schools, foundations and specialty training organisation.

**Participants** All UK medical students from 2002 to 2015, for a total of 97 932 participants.

**Outcome measures** Individual movements and number of movements by county of students from family home to medical school training, from medical school to foundation training and from foundation to specialty training.

**Methods** Leslie matrix, principal components analysis, Gini coefficient, $\chi^2$ test, generalised linear models and variable selection methods were employed to explore the different facets of students' and junior doctors' movements from the family home to medical school and for the full pathway (from family home to specialty training).

**Results** The majority of the movements between the different stages of the full pathway were restricted to a distance of up to 50 km; although the proportion of movements changed from year-to-year, with longer movements during 2007–2008. At the individual level, ethnicity, socioeconomic class of the parent(s) and the deprivation score of the family home region were found to be the most important factors associated with the length of the movements from the family home to medical school. Similar results were found when movements were aggregated at the county level, with the addition of factors such as gender and qualification at entry (to medical school) being statistically associated with the number of new entrant students moving between counties.

**Conclusion** Our findings show that while future doctors do not move far from their family home or training location, this pattern is not homogeneous over time. Distances are influenced by demographics, socioeconomic status and deprivation. These results may contribute in designing interventions aimed at solving the chronic problems of maldistribution and underdoctoring in the UK.

## INTRODUCTION

The UK National Health Service (NHS) with its 1.5 million employees in England,[1] 160 000

### STRENGTHS AND LIMITATIONS OF THIS STUDY

⇒ Use of a suite of statistical tools instead of isolated tests to ascertain significant movement scale and explanatory factors.
⇒ First use of UK Medical Education Database full data from 2002 to 2015.
⇒ Full pathway available for only a small proportion of doctors.
⇒ Additional socioeconomic explanatory factors and policy interventions will need to be considered.

in Scotland,[2] 106 000 in Wales[3] and with 73 000 employees in the health and social care in Northern Ireland,[4] is the fifth largest employer in the world.[5] The ratio between a number of doctors (general practitioners (GPs) and specialists) and the served population is approximately 2 per 1000 population. The process required to become a doctor or GP is lengthy and includes multiple training stages: medical school (4–6 years); foundation training (2 years); and specialty training (3–8 years).[6] The distribution of the doctors to population served in the UK is unequal (as shown for GPs in the past[7]), despite the sustained increase in doctor numbers in the last few years.[1] However, the shortage of doctors is unlikely to be solved in the short-term due to the structural and demographic characteristics of the underdoctored locations.[8] Therefore, before any intervention is taken it is essential to understand the patterns in geographical movements of doctors and which factors are behind them. In fact, the potential maldistribution of doctors in the UK is contingent on the spatial mobility of medical professionals, specifically in terms of their divergence from their geographical origins or training locations. If doctors predominantly remain within close proximity to their hometowns or training institutions,

the resultant geographical distribution of healthcare providers is likely to mirror the demographic characteristics of medical school entrants and the geographical locales of these educational institutions and postgraduate training facilities. This, in turn, could deviate from an ideal distribution aligned with the broader healthcare needs of the population.

Previous research had provided partial answers to the questions regarding the patterns of movements of future doctors due to limitations in the data and methods, and the coarse spatial resolution. So far, analyses of these patterns have been descriptive and hardly statistically assessed, therefore figures remain crude and not comparable.

In a study dating back to 1997, Parkhouse and Lambert[9] found that only 38.2% of doctors stayed within the region of their family home, while 42% of individuals stayed in the same area as their medical school. However, at that time, London had a larger number of medical schools compared with the rest of the UK. Hann and Gravelle reported that between the mid-1980s and 2003, the maldistribution of GPs in England and Wales increased.[7] This reduced between 1974 and 1994, although increased again in later years and by 2006 it was greater than the 1974 levels.[10]

While previous research focused on the use of simple cross-tabulation,[9] the Gini coefficient[11] and generalised linear models[12]; they were often not sufficient to disentangle the complexity of the movements of doctors. To understand the spatial scales and explanatory factors associated with the movements of entrant medical students and junior doctors, in addition to the traditional statistical analyses listed above, we have also employed a new method, the Leslie matrix (usually applied in biology). We will show that our methods provide identification of the modal movement distances (short to long), their differences over time and the demographic and socioeconomic factors associated with these scales.

## DATA AND METHODS

The data was provided by the UK Medical Education Database (UKMED) (project number UKMEDP44) and included 97 932 individuals who started medical school between 2002 and 2015. Of these, 871 had completed medical school and foundation training, and started or completed specialty training by the end of the study period (supplementary file: online supplemental informations 1 and 2). We use the terms 'students' or 'medical students' to refer to those attending a medical school, and 'junior doctors' for those in foundation or specialty training. Since the data set is only available up to 2015, the individuals for whom we have information for all three training stages are those who commenced medical school between 2002 and 2009. For all 97 932 students, the distances from the family home to medical school are available. Only individuals whose nationality was English, Welsh, Scottish or Northern Irish and whose

medical degree was awarded from a UK university were considered in this study. The distances between the family home and the medical school do not include subsequent changes in medical school. Out of 97 932 students, 5093 (5.2%) changed medical schools. The 871 individuals include only those for whom the full training pathway was available without changes. 45 additional junior doctors who, at some point, were not in training or changed location during the foundation and specialty training were excluded. The analyses included 36 medical schools and all the hospitals providing training to doctors across the UK.

This data was the full standardised data available at UKMED at that time. For each individual, the distances between each of the three career stage (family home to medical school, medical school to the foundation, foundation to specialisation) and other variables were provided (table 1). For each individual, the distances were not summed (ie, from the family home up to specialisation) and neither calculated between non-subsequent career stages (ie, from the family home to foundation or specialisation or from medical school to specialisation). We used the term 'ethnicity' as per the UKMED database, although there are ongoing debates about its appropriate use[13] in particular for medical studies.[14]

Six methods were used to investigate the movements of medical students across the UK. Four of these methods were integrated to investigate the movement distances and their patterns across the three stages and years: the Leslie model (known as a Leslie matrix), principal components analysis (PCA), Gini coefficient and $\chi^2$ test. Generalised linear models (GLMs) and variable selection methods were employed to evaluate the factors influencing the movements from the family home to medical school (which provide the largest data) at both individual and geographically aggregated levels. In practice, the first four analyses (Leslie, Gini, PCA and $\chi^2$ test) were applied to the 871 records containing the full pathways; while the Poisson GLMs and variable selection methods were applied on the whole 97 932 movements from family home to medical school.

The Leslie matrix is mostly applied to biological and ecological problems.[15] The method is used to model population changes over time, including migration.[16] It is a symmetric matrix with positive entries, where matrix cells contain the number of individuals passing from one stage to another. The matrix has been nested with distances, therefore for each stage, the number of students moving from one stage to the next is disaggregated by distance categories[17] (full method described in the supplementary file: online supplemental information 3). Since time is not integrated with space, the Leslie matrix is a simplified stage-only version. The lag distance applied is 25 km (0–25 km, 25–50 km, etc). This lag allows for the identification of patterns at finer scales not possible if using a larger lag. Only the 871 individuals who completed all three training stages, or completed

**Table 1** List of variables used in the analyses and their definitions

| Variable | Description | Categories |
|---|---|---|
| Bursary* | Whether the individual is in The University Clinical Aptitude Test (UCAT) questionnaire bursary table. | Yes<br>No |
| Distance* | Coded as the travel distance between centroids. | 0=from the county of family home to the county of the medical school; 1=from home postcode to the medical school postcode; 2=from medical school postcode to foundation training postcode; 3=from foundation training postcode to specialty training postcode. |
| Domicile county* | The county of the students' family home. | |
| Ethnicity* | Level 1 and level 2 classification of ethnic groups as in 2011 census. | Full list in supplementary file: online supplemental information 4. |
| BME* | Black and minority ethnic (BME). | 0=white<br>1=BME |
| Gender* | The sex of the student. | 0=male<br>1=female |
| Medical school county* | The county the medical school is in. | |
| Qualification* | The highest qualification the student has on entry to medical school. | 1=no formal qualification/not known; 2=level 2 and below; 3=level 3 including A levels and higher. |
| School type* | Type of school the student attended between the ages of 11 and 16. | 1=independent/private fee paying school; 2=grammar school (selective); 3=academy/secondary school; 4=further education college/sixth form college; 5=other. |
| Socioeconomic classification* | Socioeconomic background of students aged 21 and over at the start of their course, or for students under 21 the socioeconomic background of their parent, step-parent or guardian who earns the most is returned. It is based on occupation, and if the parent or guardian is retired or unemployed, this is based on their most recent occupation. | 1=higher managerial and professional occupations; 2=lower managerial and professional occupations; 3=intermediate occupation; 4=small employers and own account workers; 5=lower supervisory and technical occupations; 6=semi-routine occupations; 7=routine occupations;8=never worked and long-term unemployed; 9=not classified. |
| Deprivation† | The Index of Multiple Deprivation for England, Northern Ireland, Scotland and Wales. | 1=most deprived; 2; 3; 4; 5=least deprived. |
| Income support† | Whether the student's home has had any income support during their time at school. | Yes<br>No |
| Occupation† | Occupation of the household reference person. This is the person responsible for owning or renting or who is otherwise responsible for the accommodation. In the case of joint householders, the person with the highest income takes precedence. Where incomes are equal, the oldest person is taken as household reference. Based on the National Statistics Socioeconomic. | 1=managerial and professional occupations; 2=intermediate occupations; 3=small employers and account workers; 4=lower supervisory and technical occupations; 5=semi-routine and routine occupations. |
| Occupation individual† | Socioeconomic class occupation description of parent 1 and parent 2 from UCAT. | 1=modern professional (eg, teacher, nurse, social worker, artist, police officer (sergeant or above)); 2=clerical and intermediate (eg, secretary, call centre agent, nursing auxiliary, nursery nurse); 3=senior managers or administrators (eg, finance manager, chief executive); 4=technical and craft (eg, motor mechanic, plumber, printer, tool maker, gardener, train driver, fitter); 5=semi-routine manual and service (eg, postal/farm worker, security guard, catering/sales assistant); 6=routine manual and service (eg, HGV driver, cleaner, porter, sewing machinist, bar staff, labourer); 7=middle or junior managers (eg, office/retail/bank/restaurant/warehouse manager, publican); 8=traditional professional (eg, accountant, solicitor, medical practitioner, scientist, civil servant); 9=never worked; 10=do not know or prefer not to say. |

Source: UK Medical Education Database data dictionary.[40]
*Individual student-related variables.
†Individual student's parents-related variables.
HGV, Heavy Goods Vehicle.

the first two and started the third, were considered in the Leslie matrix construction.

The simplified stage-only Leslie matrix was analysed by employing a PCA to evaluate the most important movements' distance categories among all training stages. This approach has been proposed by Usher and Williamson[17] and the resulting eigenvectors can be compared through $\chi^2$ tests to evaluate similarities and dissimilarities in the most important movements over the years.[18] In other words, annual Leslie matrices were created and the relative first principal components (containing the largest amount of explained variance) for each year were extracted and compared in pairs by $\chi^2$ test.

To confirm equality or inequality in medical students and junior doctors (871 individuals) movement distance categories, the Gini coefficient has been calculated which has been used in similar researches.[11] The Gini coefficient varies between 0 and 1 where 0 implies complete equality, and 1 complete inequality. The coefficient is derived from the Lorenz curve. When the Gini coefficient is 0, the Lorenz curve follows the line of equality, which is a straight diagonal line. If the distribution exhibits any inequality, the Lorenz curve falls below the line of equality. The Gini coefficient is the ratio between the Lorenz curve and the equality line. The Lorenz curve is the line passing through the points of the cumulative percentage of the population (on the x-axis) and cumulative percentage of distances (on the y-axis). Finally, a Poisson GLM was applied individually on the distance (as outcome) covered by each student from the family home to medical school (therefore on the full 97932 students). Distance was assumed a count data with discrete and non-negative values. The model explanatory factors were gender, ethnicity and various variables representing the socioeconomic status of the students and their parents (full list presented in table 1). In addition, in a second Poisson GLM, the number of students moving from the county of the family home to the county of medical school was used as the outcome. Since the data is aggregated by county, the model included the distance between training and family home counties and the modal values of each explanatory variable for both the departing (family home) and arriving (medical school) counties.

Before running the individual and county-level multivariable Poisson GLMs, the selection of the explanatory variables was carried out by univariate modelling and stepwise selection[19] since all the explanatory variables are categorical. The variable selection approach included the following steps:

1. Fit univariate models with each potential exploratory variable (table 1).
2. On each univariate model, a log-likelihood ratio test (comparing with the null model) is performed using the function poisson.anova[20] in R V.4.1.0 software.
3. The explanatory variable of the univariate model was included in the multivariable model if it caused the univariate model to be significantly different (at a significant level of 0.05) from the null model.

**Table 2** Summary statistics for the three career stage movements in the full data set

| Distances | Mean (km) | SD (km) |
|---|---|---|
| From family home to medical school | 173.8 | 155.6 |
| From medical school to foundation training location | 126.8 | 226.9 |
| From foundation to specialty training location | 161.8 | 216.8 |

4. The multivariable model was then inputted in a stepwise selection method (function stepAIC in R V.4.1.0 software[21]), with both forward and backward selection; and the returned model with the lowest Akaike information criteria (AIC)[22 23] was selected.

AIC was also employed to compare the Poisson regression model with a negative binomial model, which can account for overdispersion. The former yielded a lower AIC (indicating a better fit) than the negative binomial model, and for this reason, only the Poisson regression results are shown. A post hoc test, the Tukey-Kramer test, was employed to compare pairwise differences within the variable categories. The study was performed using Strengthening the Reporting of Observational Studies in Epidemiology guidance.[24]

### Patient and public involvement

Patients or the public were not involved in the design, or conduct, or reporting, or dissemination plans of our research.

### RESULTS

The mean and SD of the distances between each stage are shown in table 2. The shorter, although more dispersed, distances are from the medical school to foundation programme training.

The PCA of the simplified stage-only Leslie matrix, applied to only the individuals for which the full pathway is known (871 individuals), found that out of 40 principal components (1 for each distance category), the first principal component (distances up to 25 km) explained 41% of the variance in the movements' distances, the second 13% (25–50 km) and the third 8% (50–75 km) (supplementary file: online supplemental information 5). The highest proportion of individuals (54%) concentrate on shorter movement distances (up to 50 km) than longer movement distances. These results have been confirmed by the value of the Gini coefficient for the Leslie matrix, which was 0.936 (supplementary file: online supplemental information 6) meaning inequality on the distribution of movements among all categories and stages, with modal distances at a short scale. These results indicate that most of the individuals stayed close to their homes or training organisation.

The proportions of movements among the distance categories are heterogeneous in time. These differences

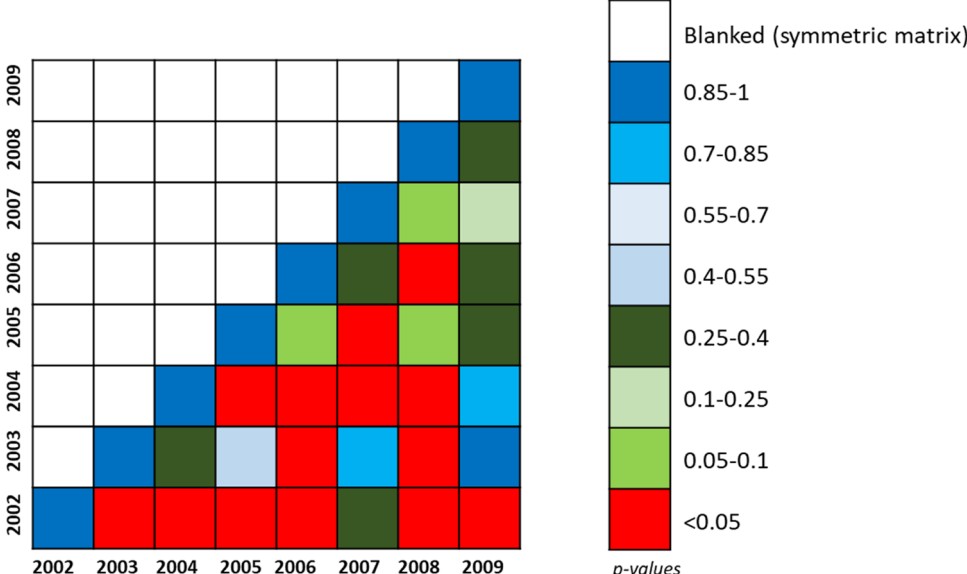

**Figure 1** X$^2$ p values from comparison of annual first components of the Leslie matrix. P values below 0.05 are shown in red.

have been tested by the $\chi^2$ test applied to paired first principal components of annual Leslie matrices (from 2002 to 2009—where 2009 is the last year for which we have a full pathway for students). Figure 1 shows that 2002 proportion of movements is significantly different from most of the rest of the years (in red) at a significant level of 0.05, apart from 2007. 2004 was similar only to 2003 and 2009; 2006 to 2005, 2007 and 2009; and 2008 to 2005, 2007 and 2009. The year that was similar to most of all the other years is 2009, which was different only to 2002. Notably, the years with the longest distances were 2007 and 2008 where the majority of individuals travelled distances up to 100–125 km (supplementary file: online supplemental information 5).

To evaluate determinants of the movement from family home to medical school at both individual and aggregated (by county) level, GLMs were employed within a variable selection framework, the latter aimed at selecting the model with the lowest AIC or in other words the best model fitting.

At the individual level, the best-fitting model of the distances travelled by students was obtained with ethnicity subgroup (level 2 information), student's parents' occupation and deprivation (table 3). This model is significantly different (in terms of reduction of residual deviance) from any model based on the three individual variables and their combinations (analysis of variance test). All the classes of deprivation and parent occupation were found to be statistically significant. In terms of deprivation, compared with the most deprived, the least deprived levels (3–5) travelled a shorter distance when they moved from their family home to medical school. Compared with students with parents in managerial and professional occupation, the students with other occupations travelled shorter distances (table 3). In terms of the ethnicity variable, most of the ethnic groups travelled shorter distances than the white British, apart from the Irish, white and

black African and other white background (Gypsy or Irish traveller coefficient was not significantly different from zero).

Among students travelling shorter distances than those with white British background, a fourfold to fivefold difference was observed between those of Bangladeshi, Arab, Pakistani and other Asian backgrounds, compared with students of Chinese, other mixed and African backgrounds (travelling longer distances compared with them but shorter than white British), and 6-fold to 10-fold difference in comparison to students of white and black Caribbean, and other black backgrounds. Moreover, there was a more than 15-fold difference in comparison to students of white and Asian backgrounds, as well as those whose background was not stated when compared with Bangladeshi, Arab, Pakistani and other Asian backgrounds (table 3).

Students of Irish background travelled the longest distances compared with those of white British background.

A multiple comparison test (Tukey-Kramer test) highlighted a pairwise significant difference between white British and: African, Bangladeshi, Caribbean, Chinese, Indian, Irish, white and Asian, white and black African, white and black Caribbean, other Asian, other ethnic and other mixed background (p value<0.001). Within the group Bangladeshi, Arab, Pakistani and other Asian backgrounds, only Bangladeshi—Arab comparison was found significantly different.

The best-fitting model by county (table 4) confirms the importance of deprivation in the number of students moving from family home to medical school counties. Distance between counties was an important factor for the number of students moving from one county to the other, with medical schools having more students coming from neighbouring counties than from counties further away (negative association with distance). The rest of the most

**Table 3** Poisson generalised linear model results for individual movements

| Variable | Coefficient | SE | P value |
|---|---|---|---|
| Intercept | 5.097 | 0.005 | <0.001 |
| Occ. managerial | REF | | |
| Occ. intermediate | −0.079 | 0.002 | <0.001 |
| Occ. small employers | −0.061 | 0.002 | <0.001 |
| Occ. lower supervisory | −0.089 | 0.003 | <0.001 |
| Occ. semi-routine | −0.080 | 0.003 | <0.001 |
| Deprivation (level 1) | REF | | |
| Deprivation (level 2) | 0.003 | 0.001 | <0.001 |
| Deprivation (level 3) | −0.045 | 0.001 | <0.001 |
| Deprivation (level 4) | −0.129 | 0.001 | <0.001 |
| Deprivation (level 5) | −0.273 | 0.001 | <0.001 |
| British | REF | | |
| African | −0.071 | 0.002 | <0.001 |
| Arab | −0.375 | 0.003 | <0.001 |
| Bangladeshi | −0.427 | 0.003 | <0.001 |
| Caribbean | −0.190 | 0.005 | <0.001 |
| Chinese | −0.088 | 0.002 | <0.001 |
| Gypsy or Irish traveller | 0.074 | 0.040 | 0.061 |
| Indian | −0.281 | 0.001 | <0.001 |
| Irish | 0.239 | 0.001 | <0.001 |
| Pakistani | −0.356 | 0.001 | <0.001 |
| White and Asian | −0.023 | 0.002 | <0.001 |
| White and black African | 0.051 | 0.004 | <0.001 |
| White and black Caribbean | −0.055 | 0.004 | <0.001 |
| Other Asian background | −0.328 | 0.001 | <0.001 |
| Other black background | −0.044 | 0.007 | <0.001 |
| Other mixed background | −0.095 | 0.002 | <0.001 |
| Other white background | 0.027 | 0.001 | <0.001 |
| Other ethnic group | −0.283 | 0.002 | <0.001 |
| Not stated | −0.015 | 0.006 | 0.006 |

Occ, occupation of the household reference; REF, reference category.

significant variables were all associated with the departing county: black and minority ethnic (BME), female students, highest qualification at entry and socio-economic classification (students socioeconomic) (table 4). In particular, counties with higher proportions of BME or with higher deprivation or socioeconomic status were associated with fewer individuals moving. While counties with a larger proportion of female students or higher qualifications at entry were associated with a larger number of movements.

## DISCUSSION

This research found that the largest proportion of individuals travelled within 25 km to the next stage of professional development, with most of the movements being statistically significant at less than 50 km, although not for every year under study. In fact, in the years of the financial crisis (2007–2008)[25] individuals within the three stages travelled further (up to 125 km). Although this study was not aimed at investigating the effect of the financial crisis on UK medical students' movements, it is certainly the first study proposing this hypothesis. A study on medical students in Spain reported that students focused more on career prospects than lifestyle in response to the economic crisis,[26] when choosing their specialty. If this was the reason for longer movements by UK medical students and junior doctors it is something that will need to be ascertained.

The up-to 50 km modal distance travelled by UK-born medical students and junior doctors for all the years between 2002 and 2009 is reported here for the first time.

**Table 4** Poisson generalised linear model results for county-aggregated data

| Variable | Estimate | SE | P value |
|---|---|---|---|
| Intercept | −0.963 | 0.068 | <0.001 |
| Distance between counties | −0.006 | 0.000 | <0.001 |
| BME departing county (no) | REF | | |
| BME departing county (yes) | −1.204 | 0.042 | <0.001 |
| Gender departing county (male) | REF | | |
| Gender departing county (female) | 2.691 | 0.029 | <0.001 |
| Qualification departing county (below level 3/not known) | REF | | |
| Qualification departing county (level 3) | 0.085 | 0.004 | <0.001 |
| Deprivation departing county (level 1) | REF | | |
| Deprivation departing county (above level 1) | −0.454 | 0.005 | <0.001 |
| Deprivation arriving county (level 1) | REF | | |
| Deprivation arriving county (above level 1) | −0.483 | 0.006 | <0.001 |
| Students socioeconomic (higher managerial) | REF | | |
| Students socioeconomic (lower and below) | −0.183 | 0.021 | <0.001 |

arriving, county with the medical school; BME, black and minority ethnic ; departing, county of family home.

A 2019 GMC descriptive report that focused on doctors who gained their UK Primary Medical Qualification between 2012 and 2018 found that almost a quarter (23%) of doctors in 2019 were working within 10 miles (16 km) of where they studied at medical school.[27] However, one-third of graduates moved more than 100 miles.

Despite the absence of distance information, a previous study (on a restricted number of doctors and obtained by questionnaire) reported that, between 1974 and 2008, 36% of the students entered a medical school that was in the same region as their family home, and 48% undertook specialist training in the same region as their medical school.[28] A more recent observational study that focused on doctors graduating between 2012 and 2014, found that the majority of doctors trained in foundation schools that were close to their family homes.[29]

We found strong inequality in the movements—for example, the presence of dominant movement distance categories compared with others. A previous study using the Gini coefficient applied to movements for general practitioners between 1974 and 2003[7] found greater equality (Gini coefficient between 0.4 and 0.5) than the present research. A direct comparison between the two studies cannot be done, but the increase in medical student number supply since 1974[7] could have caused a more pronounced movement inequality (as we found) than previously.

The main factors associated with students' movement distances and student counts from family home to medical school are very similar at both individual and aggregated—by county—levels. The most important factors were socioeconomic status, deprivation score and ethnicity (BME, ethnicity group and subgroups). In our analyses, increased deprivation was found to be associated with both fewer doctors moving and shorter distances

moved, while more students with higher degrees at entry to medical school or with parents with professional occupation tended to move further away. In previous literature, primary care trusts in England were seen to be attractive if there is less crime in the area, or better schools and opportunities.[10] Similarly, those students with parents in professional occupations were less likely to work in deprived areas compared with those in routine occupations.[30] Regarding ethnicity, one study reported that doctors from non-white ethnic backgrounds were more likely to train at foundation schools closer to their family home.[29] We found the same findings in comparison to white British and in reference to medical schools; although our analyses do not account for the potential geographical clustering of ethnic minorities in the UK[31] which can potentially bias the results. Personal factors such as ethnicity, education and gender were all significant contributing factors to the movement of doctors at both individual and county levels.[32]

Consideration of the structural elements of our societies, such as deprivation and demography (eg, ethnicity), will require different type, spatial scale and economic effort to improve the distribution of doctors in the UK. The NHS proposed a 'rural weighting' payment in England which would aim to attract doctors and GPs to rural areas with a financial incentive.[33 34] This may be equally applicable in areas of high deprivation. Payments of this type could also be increased in line with the length of service thereby increasing retention of medical practitioners within rural or deprived areas. Another potential solution is expanding widening access to medical schools and creating new medical schools in areas underserved given the tendency of the students to stay local to their training institution.[35] In Scotland it has been found that there is a link between childhood background, as defined

by parental socioeconomic status and being in a more remote residence, and the population GPs are likely to serve.[30] The authors suggested that GPs residing in more rural areas and those who experienced greater deprivation during their childhood tended to work in settings that reflected these characteristics. While this should not be presumed from these students, WHO recommendations emphasise the need to create more opportunities, such as locating educational facilities closer to rural areas, and providing incentives, such as benefits or direct payments, to medical students and doctors to serve in rural areas.[36] However, recent literature indicates that the outcomes of financial incentives are limited or unclear.[37] Family-unit and community considerations are often the predominant factors influencing doctors' movements.[38]

The main limitation of our study is the absence of historical mapping for all distances between medical schools, foundation and specialty training facilities. Currently, this information is unavailable, but it is crucial to test alternative hypotheses regarding the likelihood of movement distance in relation to the availability of training facilities in the surrounding area, weighted by the number of positions available at the training location.

Other limitations are also present. This study includes information up to 2015, which may not reflect recent changes in medical education or workforce dynamics. Students who left their medical education or moved between training locations are not included in the data set; therefore, the results are not generalisable for this group of students. We use the term 'ethnicity' as per the UKMED database, based on the categorisation adopted by the Office for National Statistics. However, this may not be appropriate due to limitations in the way this variable is defined and measured. Finally, the present study did not consider international medical graduates working in the UK, UK-born doctors working abroad or records pre-2002. Data for these doctors were not available.

A new study (NIHR134540)[39] is investigating why doctors work where they work and how this is influenced by the organisation of the healthcare system, the social and physical environment where the future doctors grow up and move as well as their individual beliefs and choices. This new study which uses mixed methods and a larger data set than any previously should increase the understanding of doctors' movements from the doctor's perspectives as well and not only from data records. We hope that our study and the ones currently active will help to identify the best interventions to improve the under-doctoring and maldistribution of doctors across the UK.

**Author affiliations**
[1]Mathematics and Statistics, Lancaster University, Lancaster, UK
[2]Lancaster Medical School, Lancaster University, Lancaster, UK
[3]General Medical Council, London, UK
[4]Centre for Health Informatics, Computing and Statistics, Lancaster Medical School, Lancaster University, Lancaster, UK
[5]School of Medical Sciences, The University of Manchester, Manchester, UK
[6]General Medical Council, Manchester, UK
[7]Lancaster Ecology and Epidemiology Group, Lancaster Medical School, Lancaster University, Lancaster, UK

**Acknowledgements** We are grateful to UKMED for the use of its data. However, UKMED has no responsibility for the analysis or interpretation of their data. The data includes information derived from that collected by the Higher Education Statistics Agency Limited ('HESA') and provided to the GMC ('HESA Data'). Source: HESA Student Records 2002/2015 Copyright Higher Education Statistics Agency Limited. The Higher Education Statistics Agency Limited makes no warranty as to the accuracy of the HESA Data, cannot accept responsibility for any inferences or conclusions derived by third parties from data or other information supplied by it.

**Contributors** CM and LS conceived and designed the study. LH, DTS, JMR, CM and LS discussed and drafted the study protocol. LH and BF screened and selected the articles. LH and BF analysed the data. LH, BF and LS interpreted the results. LH, BF and LS drafted the manuscript. All authors critically revised the manuscript, contributed to revising the manuscript and had final responsibility for the decision to submit for publication. DTS, JMR, CM and LS supervised the study. The corresponding author attests that all listed authors meet authorship criteria and that no others meeting the criteria have been omitted. DTS, JMR, CM and LS are jointly supervised this work. LS is the guarantor and accepts full responsibility for the finished work and/or the conduct of the study, had access to the data, and controlled the decision to publish.

**Funding** LS is supported by the NIHR Health and Social Care Delivery Research (HSDR) programme (NIHR134540) and North West Cancer Research UK (LI2021SEDDA). The views expressed are those of the author(s) and not necessarily those of the NIHR or the Department of Health and Social Care.

**Competing interests** CM is a medical director and director of education and standards at the General Medical Council. Views expressed in this article do not necessarily reflect those of the General Medical Council.

**Patient and public involvement** Patients and/or the public were not involved in the design, or conduct, or reporting, or dissemination plans of this research.

**Patient consent for publication** Not applicable.

**Ethics approval** The Medical Schools Council agreed that all approved applications for research projects using data exclusively held by UKMED would meet the criteria for a blanket exemption from the need to apply for ethics approval that would be recognised by ethics committees relevant to the UK medical schools. A letter was written from QMUL Ethics of Research Committee on behalf of all UK medical schools to confirm ethics exemption (available at chrome-extension://efaidnbmnnnibpcajpcglclefindmkaj/https://www.ukmed.ac.uk/documents/UKMED_research_projects_ethics_exemption.pdf).

**Provenance and peer review** Not commissioned; externally peer reviewed.

**Data availability statement** No data are available. The data for this research was accessed through the cloud from the UK Medical Education Database (UKMED), specifically from the UKMEDP044 extract generated on 1 June 2018. The availability of the data used in this study is through an application process to UKMED, and more information can be found on their website: https://www.ukmed.ac.uk/. Approved for publication on 30 June 2023.

**ORCID iDs**
Ben Fleet http://orcid.org/0000-0002-8832-8531
Daniel Thomas Smith http://orcid.org/0000-0003-1215-5811
Jonathan M Read http://orcid.org/0000-0002-9697-0962
Luigi Sedda http://orcid.org/0000-0002-9271-6596

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
