## [Reviewer comments · BMJ Open]

This paper was submitted to a another journal from BMJ but declined for publication following peer review. The authors addressed the reviewers' comments and submitted the revised paper to BMJ Open. The paper was subsequently accepted for publication at BMJ Open.

ARTICLE DETAILS

TITLE (PROVISIONAL)	Determining the distance patterns in the movements of future doctors in UK between 2002 and 2015: a retrospective cohort study
AUTHORS	Hitchings, Lucy; Fleet, Ben; Smith, Daniel; Read, Jonathan; Melville, Colin; Sedda , Luigi

VERSION 1 – REVIEW

REVIEWER	Hogenbirk, John Laurentian University, Centre for Rural and Northern Health Research
REVIEW RETURNED	18-Sep-2023

GENERAL COMMENTS	Abstract The abstract is generally well-written. Additional information and suggestions for revision may help improve readability. Line 58: While the study does address physician movement, the study does not specifically address “spatial patterns” per se. Please rephrase. The link between the Aim and the Methods needs to be made more explicit. Details are needed on the nature of the data, and an explicit connection to the specific statistical technique will help reader to better understand the approach and results. Given the novelty of the Leslie model approach, consider including a brief description of the data used to populate the matrix and the features of the matrix. Additional clarity in the Results will also help. The authors write that certain factors were significantly associated with physician movement, but do not specify whether the association is positive or negative. For example, the authors should consider adding a sentence such as “Factors X, Y and Z were associated with more physicians moving greater distances whereas P, Q and O were associated with more physicians moving shorter distances.” If
--

space permits, include the distinction between individual and aggregate findings.

If additional details are added to the Results, then only a few revisions need to be made to the Conclusions. For instance, consider specifying the positive and negative associations between physician-distance counts but only for the strongest explanatory factors.

Introduction

The introduction is well-written, contains salient references, and explains the novel features of the study. The authors should consider adding a few clauses or sentences to explicitly link movement and maldistribution of physicians. A sentence similar to the following might help. "If physicians do not move far from where they grew up or where they are trained, then maldistribution of physicians in the UK will reflect the hometowns of students who are admitted into medical school and the location of these medical schools or postgraduate training sites rather than the distribution and care needs of the population."

Line 103 makes reference to the use of the phrase "confusion matrix". "Cross tabulation" may be a better phrase given that a "confusion matrix" implies misclassification. The fact that a physician's hometown and clinical medical school differ in location is not a misclassification.

Methods

As noted in the comments on the Abstract, the link between the Purpose and the Methods needs to be made more explicit. In addition, some material in the Results section are better placed in the Methods section. The authors should also make reference to their use of the STROBE checklist and cite von Elm et al.

Line 117: states: "For each individual, the distances between each career stage... and other variables were provided." However, line 159 of the Results states that only 871 have data on the full path. Please add text to the Methods section to explain this subset.

It is also not clear whether distances were available from home to foundational training location or from home to specialization training location. Based on the description, it seems that distances were only available between each stages. Is this interpretation correct?

Lines 122-126: it should suffice here to list the methods and defer the full explanation to the following paragraphs.

Line 124: regardless of whether this paragraph is revised, please clarify what is meant by "scales" in the phrase "movements scales". I thought that the data were counts of physician-distances by medical education stage.

Line 127: My understanding of Leslie matrices is that the matrix can take one of two basic forms: (1) tracking a closed cohort through time (e.g., people born in 1995, surviving to age 1, 2, ...); or (2) a cross-sectional snapshot of the population at one time (e.g., UK population in 2023, by 1 year age class).

Type 1 seems most appropriate to this study, albeit with the matrix having more than one cohort (i.e., students starting medical school in years 2002-2009).

Was there a separate matrix for each cohort? Or were all cohorts combined? Nested?

Is it correct to assume that all students have completed all stages in their medical education pathway? Please confirm. Else provide percentages at each stage (by cohort).

Line 130: Consider removing "(in a sort of pyramidal spread)". I found that this clause confusing. Consider replacing with more fulsome description on how the Leslie matrices were created. Note, if this section starts to get too wordy, then consider creating a supplement with the full description.

My understanding is that the table of students by education location (in which distance groups are nested) were used to derive the probability of movement to the next education location—this would be equivalent to the survival probability (S) of a typical Leslie matrix (see example below). If all cohorts were combined then the fertility probability (F) of a typical Leslie matrix would not be applicable and these cells in the matrix would be zero. If cohorts were treated separately, then my guess is that the fertility probability would be applied to the first stage only (i.e., incoming medical students). A simplified Leslie matrix is shown below for the 3 stages—all cohorts combined. If I understand correctly, then the 25 km distance groups would be nested within the non-zero cells (specifically S0 and S1). Please confirm if this understanding is correct and add the salient details to the Methods (or Supplement).

$$\begin{bmatrix} & F1 & F2 & F3 \\ S0 & 0 & 0 & \\ 0 & S1 & 0 & \end{bmatrix}$$

Line 130: Given that the matrix comprises counts of physician-distances, it is more accurate (albeit more wordy) to say "from home to medical school, from medical school to foundational training location, from foundation to specialization training location".

Questions: Can learners change locations during medical school, foundational training or specialty training? If yes, then how common is this changes? If yes and if this is common, then how were these changes captured in the data?

Line 141: Please clarify the data that were used to calculate the Gini coefficient and please explain how the line of equality was derived.

Was the coefficient and line derived for all movement stages or just for home to medical school?

Given that schools and training locations tend to be located in urban areas and given that the population is predominantly located in urban areas, then presumably the line of equality should match the distribution of distances from the general population to medical school.

In contrast, distances from medical school to foundation location would reflect a different distribution: distances between medical schools and foundation training locations weighted by the number of learners at medical schools and the number of positions available at the foundation training location. A similar argument would apply to the expected distribution of distances between foundation and specialization training locations.

The caption and axes titles for Figure 1 may need additional information on how the Lorenz curve and line of equality were derived.

Line 144, presumably the logistic Poisson GLM was applied to the “count of physician-distances” and not to the “distances” per se. Was this model applied to each mobility’s stage or just to the home to medical school mobility stage? Was any consideration given to the hierarchical nature of some of the data (Specifically, individual level data, county level data)? A generalized linear mixed model might be a more appropriate statistical technique for dealing with hierarchical data.

From the Methods it seems that a second Poisson analysis was applied to aggregated data. Please clarify.

Trivial, but if the Poisson analysis was the final analysis, then why is ANOVA mentioned as a separate analysis (here and in the Results)? Are the authors referring to the ANOVA component of the Poisson GLM? And that they used backwards-forward selection on the Poisson model? Please clarify.

Results

The results section is generally well-written but could use minor revision to improve clarity.

Line 159: As noted previously, the subset of 871 individuals needs to be explained in the methods section and removed from the Results. For example, in the Methods write something like “Of 97,932 students, only 871 have data on the full path from family home to specialization.”

Lines 159-136: Remove these lines and move salient information into the Methods section. Line 164 becomes the first line in the Results section and it can be modified slightly to read “The analyses of the Leslie matrix, applied only to the 871 students for which the full pathway is known, found that...”

Consider promoting supplemental tables A2 and A3 to become full table in the manuscript.

Figure 1 can become a supplement.

Lines 195-196: Sentence is incomplete...please revise.

Line 197: Is the model fitting “distances” or “counts of physician-distances”, please clarify here and elsewhere.

Lines 198-199: I strongly suggest using the brief variable name (a short descriptive name) rather than the abbreviation. For example, “occupation category” rather than NSSEC, and “deprivation quintile” rather than IMD. Although the full explanation of the different variables and their categories belongs in Table 1, it would help the reader’s if a short descriptive name was used rather than referring the reader back and forth to Table 1.

Line 204: Please keep using the phrase “shorter distance” rather than “lower distance”. Please check and ensure consistency when referring to “distances”, “movements”, etc. As noted previously, technically the data are counts of students/physicians who have travelled specific distances (i.e., counts of physician-distances) and not distances per se. You can interpret the “count of physician-distances” as “distances” in the Discussion (with the appropriate caveat), but in the Results please use the precise term.

Discussion

The discussion is generally well-written with the salient material presented.

Line 229: trivial, ascertain should be ascertained.

Lines 230-231: Sentence is somewhat awkward and could do with revision (and a spelling check). Consider something like “The finding that the majority of UK-born [spelling] physicians in training travelled less than 50 km during 2002-2006 and in 2008 is reported here for the first time in the literature.”

Lines 231-233: Further clarification of what is reported in reference 23 would help. Did reference 23 speak specifically to foundational and/or specialty training location or was practice location included?

Line 249: consider replacing “lower number of doctors moving” with “fewer doctors moving”. And add more” to “more students with higher degrees at entry...”

Lines 251-254: In summarizing findings from reference 26, is it correct to say that the students were “working” in deprived areas? Perhaps “studying” or “training” would be a better word choice. Are physicians still called “students” during foundational or specialty training? I ask because typically, the medical learner is a “doctor” and no longer a student once they finish their MD, MBBS, or equivalent.

Line 259: Consider revising this sentence as “that the geographic distribution of ethnic minorities in UK is clustered (27)”

Lines 267-269: Please add more detail to clarify how access to medical care, childhood background and the class of population that the GP is likely to serve is connected.

As noted earlier, if physicians stay close to home or stay close to where they trained, then the maldistribution of physicians in the UK will reflect the applicant pool/admissions process, and the location of medical schools or postgraduate training sites rather than the distribution and care needs of the population. There is considerable evidence from the UK and internationally supporting the selection of students from underserved areas and the location of training sites in underserved areas as one approach to addressing this maldistribution. See, for example WHO 2021
<https://apps.who.int/iris/handle/10665/341139>

I would also mention that financial incentives have had mostly limited and short-term success in recruiting physicians (or healthcare workers in general) to underserved areas and retention tends to be poor.

Line 270-271: This sentence is awkward and requires revision. Consider “the present study did not consider international medical graduates who work in the UK, or UK-born [spelling] doctors working abroad: data for these physicians were not available.”

Line 274: “individual beliefs”[plural] and “The proposed study...” rather than “This study...”.

Line 285: Acknowledgements. Should “I am grateful...” be revised to read “We are grateful...” ?

References

The range of references seems appropriate to the topics covered in the manuscript. However, there are a number of references with incomplete information or have incorrect formatting: journal titles are missing (e.g., #8, 14, 15); different abbreviations or different names are used for journals; citation information is missing (e.g., #28) and capitalization is inconsistently or incorrectly applied to proper nouns (e.g. #8 “british doctors”) and to journal names/abbreviations. Citations for IBM SPSS and STROBE should be added.

Tables and Figures

Table 1: Additional information is needed. It would be best not use variable names in the first column given that variable names only have meaning for the research team. Please use brief descriptive names in the first column (Variable) and ensure that the Description has the full name (not the variable name) and abbreviation. The four distance measures can be combined as indicated below. Similarly for Ethnicity, Income, Occupation (NSSEC?) and please group similar variables together (e.g., Ethnicity with BME, Occupations-NSSEC).

Variable	Description
Distances	Road distances (? straight-line distances ?) from: 0-home county to medical school county 1-home to medical school 2-medical school to foundation training location 3-foundation training location to specialty training location.

Distance: is this road distance based on exact addresses, based on postal code centroids, or straight-line Euclidean distance to specific locations? or to centroids?

Please define HESA and SEC.

NSSEC when first mentioned applies to whom, the student? The student’s parents? Please clarify.

“Quintile” needs additional information in the name. Consider labelling this variable “Deprivation”.

SEC: how does this variable differentiate between the student’s classification when the student is over 21 or the student’s parents classification (when the student is under 21 years of age)?

	Occupation 1 and 2: presumably this is the NSSEC value for the parents. Please clarify. School type: please list the type of school categories. Please provide a reference for these variables. If variables are fully defined in the UKMED data dictionary, then it should suffice to reference the data dictionary. Figure 1: The caption and axes titles may need additional information on how the Lorenz curve and line of equality were derived. Consider moving this figure to the supplements. Figure 2: Is the table supposed to be asymmetrical? I ask because if you read the table from top to bottom, then you do not get the results that the authors describe. If the upper left diagonal was deliberately removed, then I suggest using white (blank) cells and explaining this feature in the figure caption. Figure A3: Trivial, but please replace (up) and (down) with (top) and (bottom) or label as (a) and (b). Both graphs are “up” from the caption. Please correct the spacing in the caption. Tables A2 & A3: Consider moving these tables to the body of the manuscript. Please use meaningful labels rather than the variable names. Otherwise you force the reader to go back and forth between Table 1 and the Results tables. “Estimate” should be relabeled as “Coefficient” I assume that this the raw coefficient (i.e., not standardized). Please clarify. Consider adding standardized coefficients to help the reader assess the relative importance of each variable. Formatting of values in the table should reflect common usage. While I appreciate the use of scientific notation and the 4 significant digits, most readers would benefit from seeing the values in straight forward decimal format with 3 digits (e.g., -0.963, -0.006). Formatting of p-values should reflect common usage: $p < 0.001$ (< 0.0001) for the smallest values should suffice.
--	--

REVIEWER	Holdren, Sarah The University of North Carolina at Chapel Hill School of Medicine
REVIEW RETURNED	19-Sep-2023

GENERAL COMMENTS	Thank you for allowing me to review this interesting work. The description of the study is excellent and fills a need in the literature. There are only a few minor typos that will need edited during copy editing.
--

REVIEWER	Foldager, Leslie Aarhus Universitet, Dept. of Animal Science
-----------------	---

GENERAL COMMENTS

I have been asked to provide a statistical review and will therefore focus on this. If I stumble over something else, then I will of course mention this, but I have not read the full manuscript thoroughly. This is also the reason for some of the N/A choices in the review checklist above.

It appears to me as an interesting and important investigation. I think parts of the manuscript could be improved a bit and at least some of the analyses are not described clearly enough, leaving doubts as to what was actually done. I am sure this can be handled without re-doing everything though some consideration may be needed regarding the relatively many categories of distance. Would it be possible to include more students by omitting some variables? or is it the information on the actual movement (and distance), which is missing? I really miss some explanation why it was impossible to get individual data, at all, from the last six years (2010-15).

This study aimed to understand movements (or distance of movements) and spatial patterns of medical student throughout their training. The study found that the students do not move far but with an inhomogeneous pattern over time and within individual characteristics. Statistical analyses were integrating different methods and applied on both individual and aggregated movements. Six methods have been used for various parts of the analyses: 1) Leslie model (Leslie matrix), 2) PCA, 3) Gini coefficient, 4) chi-squared test, 5) GLM, and 6) ANOVA. The Leslie matrix was analysed by use of PCA and comparing resulting eigenvectors through chi-squared test. Moreover, Gini coefficients were calculated to confirm presence of dominant distance categories. The GLM concerned (at least) the distances travelled by the individual students. Was it using the distance as such as response? (not clear to me from L144-145)

The study was based on registry data from the UK Medical Education Database (UKMED), including 97,932 students who started medical school (one of 36 schools) between 2002 and 2015. Distances were available for 3 stages on the individual level: 1) family home to medical school, 2) medical school to foundation, and 3) foundation to specialisation. In addition, distance from the county of the family home to the county of the medical school was available, and two variables identifying these counties. Moreover, several variables related to gender, ethnicity, socio-economy, etc. (listed in Table 1). Re-reading my comments I wonder if only distances from family home to medical school was analysed? or was this just in relation to the county-based analysis?

Table 1: I do not think it is totally clear in all cases whether the variable concerns the student or the student's parents/family. The variable 'Gender' would be more clearly denoted 'Female'.

L127-128: Just wondering that you say the method was first developed in the 1960s but the reference (no. 15) is from 1945!? The reference is also not complete in the ref. list ... it is from Biometrika. Generally, the reference list is rather messy and incomplete!

L144: What is a 'logistic Poisson' GLM? The Poisson family does to my knowledge allow for a log, identity or sqrt as link function. Did you run a quasi-Poisson model? Moreover, you say 'distances' in plural ... so did each student contribute with more than one observation to this model? In that case how about correlations among repeated observations from the same student?

L147: To me it sounds like both this model and the model in L144 might be over-dispersed, and this may be a reason for choosing a quasi-Poisson model. But why a logistic link as opposed to say a negative binomial model with log link? I miss some more description of the considerations behind the model choices.

L150-151: It seems counterintuitive to talk about 'univariate modelling' in the context of backward-forward selection, which would normally be used to build multivariable models. Moreover, I would call it 'forward selection' and 'backward elimination' (as you are not selecting variables but rather deselecting, i.e., eliminating variables, in the backward part). The process is so well known that I do not think a reference is needed – however, you need to state your thresholds. Also, if you start from some model with several variables (which could be one way to go), then state what the initial model include. If you start with the null model (and adds to this in a forward manner, followed by backward elimination) than state this. How about interactions?

L151-152: To me ANOVA means that a normal distribution of residuals has been assumed (i.e., a normal linear model) but you have just described Poisson models so this (univariate?) testing of explanatory variables by ANOVA is by no means clear to me. Or did you perhaps mean that the ANOVA was used for the county-based analysis? This is however not what one will understand from 'the same model' in L147.

L153: So, you did not test the increase in variance mentioned in L152 or did you just look at the AIC and included an extra variable if the AIC decreased? Or how is this to be interpreted? Moreover, here it sounds as if you choose among several models, but this is not as such the case if you used a "forward selection backward elimination" procedure ... rather this is a model building procedure. AIC can be used for this and sometimes is but then you are not lining the models up and choosing. AIC is also sometimes used for decision when models are not nested – but then it would not be a forward/backward process.

L150-153: All together, I find the description unclear and I am not sure what you actually did.

L159-161: How did you determine that the 871 students are representative for the full dataset? Did you just look at the histograms and then by eye decide for yourself that this looks fine? Wouldn't it be possible to test? For example, how about splitting the data into two (1) the 871 and (2) the rest, and then formally test if the distributions are not different (perhaps by non-parametric statistics if it does not make sense to do a parametric model)? Do you have any explanation why paths are only complete for (some) students from years 2002-2009? Why are the last six years less complete?

L167: If 54% of the 871 students happens to be in the 0-25 and 25-50 km categories, then there must be some of the other 38 categories that include very few (perhaps even zero) students!? Splitting this on years will obviously give even fewer in each category. How robust are these analyses to zero cells or cells with very few expected counts? Do the categories have to be with same lag (25 km) or would collapsing categories be a way to avoid having (too?) few students in some categories?

L170: Here and a couple of other places you use the words 'equity' and 'inequity'. I guess (most of) you have English as your native language, so I almost do not dare questioning this. Nevertheless, did you want this to be 'equity' and not "equality"? In Fig. 1 you use 'equality' and 'inequality' so at least you leave some unclarity for us non-native English speakers.

L182-186 and Fig. 2: Do you not have a multiple comparison problem here that should be accounted for? Perhaps you should mention (to make this clear to all) that those in red are interpreted as having significantly different proportion(s?) of short movements. Supposedly the matrix should be symmetric and thus, if presented, the upper left triangle should have the same colours as the lower right triangle. Wouldn't it be better to leave out one of the triangles? Moreover, I suppose comparing a year with itself is not meaningful so the diagonal should also be left empty. In the figure text you could mention what you understand by 'first components'. Likewise, it is not clear what you understand by 'short movements' (L182). Is it only the 0-25 category? or is it both 0-25 and 25-50? or ...?

Table A2: I would think that the overall test (p-value) for the variable would make more sense to present and then by letters indicate which levels are different/not different when compared pairwise ... after relevant adjustment for multiple comparison, of course. Regarding small values, either you have to superscript '-16' (and other equivalent values) or use the E-notation, e.g., $<2e-16$ (personally I prefer having a white space between '<' and the number, i.e., $< 2e-16$, but this is just a preference). Another personal preference is to present standard errors with 1 more decimal than the means. 'Quintile' ??? quintile of what?

L204 and L206: It is not clear what you mean by 'lower distances' and 'longer distances' ... where is the cut?

Table A3: A number like -9.629×10^{-1} is better presented as -0.9629 (I would probably say that 3 decimals, i.e., -0.963, is sufficient) and correspondingly, 5.862×10^{-2} as 0.5862. You could perhaps write e.g., < 0.0001 instead of 3.175×10^{-5} . At least remove the prefix 0's and as mentioned for Table A2, use superscript. It is not super clear what is meant by 'Quintile Depart' and 'Quintile Arrive' ... is it a 0/1 dividing into \leq / $>$ the quintile of depart (arrive) distance? Perhaps add a footnote explaining this. 'Gender Depart' ??? What do you mean by 'statistical summary of the explanatory variables' coefficient'? Isn't it the estimate from a model? If not, what is it then? I do not think it is possible to understand this table stand-alone. Perhaps it is by consulting the methods section but a variable like 'Gender Depart' is mysterious to me ... how is this defined for the number of students leaving home _from_ a specific county? Moreover, if you do not change 'Gender' to Female (see comment to Table 1), you will need to tell

	that Gender=1 are females and Gender=0 (the reference) are males. If any of the variables are continuous, we need to know the unit for the estimate/coefficient. If it is not continuous, what is then meant by e.g., the 'Distance' variable? and what are reference categories for those that are dichotomous or categorical? L285: 'I am grateful' ... I hope all authors are grateful – shouldn't this be 'We are grateful' or 'The authors are grateful'? This would also correspond better to 'for their analysis' ... I hope UKMED bears responsibility for their own analysis though if this was what you meant.
--	--

VERSION 1 – AUTHOR RESPONSE

Reviewer: 1

Mr. John Hogenbirk, Laurentian University, Northern Ontario School of Medicine

- Abstract.

-The abstract is generally well-written. Additional information and suggestions for revision may help improve readability. Line 58: While the study does address physician movement, the study does not specifically address “spatial patterns” per se. Please rephrase.

Sentence rephrased: To determine and identify distance patterns in the movements of medical students and junior doctors between their training locations. [Lines 23-24]

- The link between the Aim and the Methods needs to be made more explicit. Details are needed on the nature of the data, and an explicit connection to the specific statistical technique will help reader to better understand the approach and results. Given the novelty of the Leslie model approach, consider including a brief description of the data used to populate the matrix and the features of the matrix.

The abstract has been modified to adhere to the journal's format. Regarding the methodology, while we acknowledge the need for additional description (as provided in the method section and supplementary file), the 300-word limit for the abstract constrained the amount that could be written for the methodology.

- Additional clarity in the Results will also help. The authors write that certain factors were significantly associated with physician movement, but do not specify whether the association is positive or negative. For example, the authors should consider adding a sentence such as “Factors X, Y and Z were associated with more physicians moving greater distances whereas P, Q and O were associated with more physicians moving shorter distances.” If space permits, include the distinction between individual and aggregate findings.

We made slight changes to the results section to improve clarity. However, due to the limited word count, we have not provided a full description of the effects of each factor. The majority of the movements between the different training stages were restricted to a distance of up to 50 km; although the proportion of movements changed from year to year, with longer movements during 2007-2008. At the individual level, ethnicity, socio-economic class of the parent(s), and the deprivation score of the family home region were found to be the most important factors associated with the length of the movements from family home to medical school. Similar results were found when movements were aggregated at the county level, with the addition of factors such as gender and qualification at entry (to medical school) being statistically associated with the number of students moving between counties. [Lines 35-42]

- If additional details are added to the Results, then only a few revisions need to be made to the Conclusions. For instance, consider specifying the positive and negative associations between

physician- distance counts but only for the strongest explanatory factors.

Once again, we apologise for not incorporating additional details into the abstract. The 300-word limit constrains our ability to add more information. Consequently, the conclusion section remains largely unchanged.

- Introduction.

- The introduction is well-written, contains salient references, and explains the novel features of the study. The authors should consider adding a few clauses or sentences to explicitly link movement and maldistribution of physicians. A sentence similar to the following might help. "If physicians do not move far from where they grew up or where they are trained, then maldistribution of physicians in the UK will reflect the hometowns of students who are admitted into medical school and the location of these medical schools or postgraduate training sites rather than the distribution and care needs of the population."

A new sentence has been added as follows: In fact, the potential maldistribution of doctors in the United Kingdom is contingent upon the spatial mobility of medical professionals, specifically in terms of their divergence from their geographic origins or training locations. If doctors predominantly remain within close proximity to their hometowns or training institutions, the resultant geographic distribution of healthcare providers is likely to mirror the demographic characteristics of medical school entrants and the geographic locales of these educational institutions and postgraduate training facilities. This, in turn, could deviate from an ideal distribution aligned with the broader healthcare needs of the population. [Lines 71-78]

- Line 103 makes reference to the use of the phrase "confusion matrix". "Cross tabulation" may be a better phrase given that a "confusion matrix" implies misclassification. The fact that a physician's hometown and clinical medical school differ in location is not a misclassification.

We appreciate the reviewer's feedback; indeed, 'cross-tabulation' is the correct term rather than 'confusion matrix,' and we have made the necessary adjustment. [Line 89]

- Methods.

- As noted in the comments on the Abstract, the link between the Purpose and the Methods needs to be made more explicit. In addition, some material in the Results section are better placed in the Methods section. The authors should also make reference to their use of the STROBE checklist and cite von Elm et al.

We have now explicitly stated the purpose of each group of methods and indicated the dataset to which they are applied. Additionally, some information previously found in the results section has been relocated to the methods. The new paragraph is as follows: Six methods were used to investigate the movements of medical students across the UK. Four of these methods were integrated to investigate the movement distances and their patterns across the three stages and years: the Leslie model (known as a Leslie matrix), principal component analysis (PCA), Gini coefficient and chi-square test. Generalized linear models (GLMs) and variable selection methods were employed to evaluate the factors influencing the movements from family home to medical school (which provide the largest data) at both individual and geographically aggregated levels. In practice, the first four analyses (Leslie, Gini, PCA and chi-square test) were applied to the 871 records containing the full pathways; while the Poisson GLMs and variable selection methods were applied on the whole 97,932 movements from family home to medical school. [Lines 122-130]

We have added a STROBE statement with relative reference: The study was performed using Strengthening the Reporting of Observational studies in Epidemiology (STROBE) guidance(24). [Lines 181-183]

- Line 117: states: "For each individual, the distances between each career stage... and other variables were provided." However, line 159 of the Results states that only 871 have data on the full

path. Please add text to the Methods section to explain this subset.

We have now provided an explanation for this subset at the beginning of the Data and Methods section: The data was provided from the UK Medical Education Database (UKMED) (project number UKMEDP44) and included 97,932 individuals who started medical school between 2002 and 2015. Of these, 871 had completed medical school and foundation training, and started or completed specialty training by the end of the study period (Supplementary File: Supplementary Information 1 and 2). We use the terms 'students' or 'medical students' to refer to those attending a medical school, and 'junior doctors' for those in foundation or specialty training. Since the dataset is only available up to 2015, the individuals for whom we have information for all three training stages are those who commenced medical school between 2002 and 2009. For all 97,932 students, the distances from the family home to medical school are available. [Lines 99-107]

- It is also not clear whether distances were available from home to foundational training location or from home to specialization training location. Based on the description, it seems that distances were only available between each stages. Is this interpretation correct?

Yes this is correct, distances are only available between each stage. We have added the following sentence to improve clarity: For each individual, the distances between each career stage (family home to medical school, medical school to foundation, foundation to specialisation) and other variables were provided (Table 1). For each individual, the distances were not summed (i.e., from the family home up to specialisation) and neither calculated between non-subsequent career stages (i.e., from the family home to foundation or specialisation or from medical school to specialisation). [Lines 115-120]

- Lines 122-126: it should suffice here to list the methods and defer the full explanation to the following paragraphs.

In response to the aforementioned comment, the section has been rewritten to clarify how and when these methods were used. Full explanations have been deferred to the subsequent paragraphs: Six methods were used to investigate the movements of medical students across the UK. Four of these methods were integrated to investigate the movement distances and their patterns across the three stages and years: the Leslie model (known as a Leslie matrix), principal component analysis (PCA), Gini coefficient and chi-square test. Generalized linear models (GLMs) and variable selection methods were employed to evaluate the factors influencing the movements from family home to medical school (which provide the largest data) at both individual and geographically aggregated levels. In practice, the first four analyses (Leslie, Gini, PCA and chi-square test) were applied to the 871 records containing the full pathways; while the Poisson GLMs and variable selection methods were applied on the whole 97,932 movements from family home to medical school. [Lines 122-130]

- Line 124: regardless of whether this paragraph is revised, please clarify what is meant by "scales" in the phrase "movements scales". I thought that the data were counts of physician-distances by medical education stage.

The sentence has been removed from the methods; however, it remains in the introduction, where it has been modified to enhance clarity: We will show that our methods provide identification of the modal movement distances (short to long), their differences over time and the demographic and socio-economic factors associated with these scales. [Lines 93-95]

- Line 127: My understanding of Leslie matrices is that the matrix can take one of two basic forms: (1) tracking a closed cohort through time (e.g., people born in 1995, surviving to age 1, 2, ...); or (2) a cross-sectional snapshot of the population at one time (e.g., UK population in 2023, by 1 year age class). Type 1 seems most appropriate to this study, albeit with the matrix having more than one cohort (i.e., students starting medical school in years 2002-2009). Was there a separate matrix for each cohort? Or were all cohorts combined? Nested? Is it correct to assume that all students have completed all stages in their medical education pathway? Please confirm.

We utilised both a cross-sectional snapshot Leslie matrix and an annual, cohort-based Leslie matrix, to analyse changes over the years. For this part of the analyses, our focus was exclusively on students who completed their education pathway. We have revised the text to convey these points more clearly: The Leslie matrix is mostly applied to biological and ecological problems (15). The

method is used to model population changes over time, including migration (16). It is a symmetric matrix with positive entries, where matrix cells contain the number of individuals passing from one stage to another. In this context, the stages are the education stages (medical school, foundation, specialisation). The matrix has been nested with distances, therefore for each stage, the number of students moving from one stage to the next is disaggregated by distance categories (17) (full method described in the Supplementary File: Supplementary Information 3). Since time is not integrated with space, the Leslie matrix is a simplified stage-only version. The lag distance applied is 25 km (0-25 km, 25-50 km, etc...). This lag allows for the identification of patterns at finer scales not possible if using a larger lag. Only the 871 individuals who completed all three training stages, were considered in the Leslie matrix construction.

The simplified stage-only Leslie matrix was analysed by employing a principal component analysis to evaluate the most important movements' distance categories among all training stages. This approach has been proposed by Usher and Williamson (17) and the resulting eigenvectors can be compared through chi-square tests to evaluate similarities and dissimilarities in the most important movements over years (18). In other words, annual Leslie matrices were created, and the relative first principal components (containing the largest amount of explained variance) for each year were extracted and compared in pairs by chi-square test. [Lines 131-147]

Else provide percentages at each stage (by cohort).

Provided in Supplementary Information 1 and 2:

Supplementary information 1: Table S1. Cohort of medical students and junior doctors considered in the study. Foundation year one (F1) and Specialty year one (S1) numbers are referred to individuals that started the medical school between 2002 and 2015.

Year	Number medical students starting medical school	Number of junior doctors in F1	Number of junior doctors in S1
2002	6640	0	0
2003	6824	0	0
2004	7190	0	0
2005	7201	0	0
2006	7145	0	0
2007	7042	0	0
2008	7193	0	0
2009	7100	4487	0
2010	7130	5331	0
2011	7193	5684	137
2012	6936	6494	309
2013	6751	6378	236
2014	6810	6752	114
2015	6777	6597	75

Totals	97932	41723	871
--------	-------	-------	-----

Supplementary information 2: Table S2. Number of junior doctors starting S1 between 2011 and 2015 (second row) and contribution by medical school year.

Year junior doctors started S1	2011	2012	2013	2014	2015
Number of junior doctors in S1	137	309	236	114	75
Proportion of junior doctors in S1 (second row) that started medical school in 2002	14.39	3.39	2.65	2.75	2.78
Proportion of junior doctors in S1 (second row) that started medical school in 2003	75.76	29.49	11.95	14.68	6.94
Proportion of junior doctors in S1 (second row) that started medical school in 2004	6.82	58.98	18.14	28.44	8.33
Proportion of junior doctors in S1 (second row) that started medical school in 2005	2.27	7.12	63.72	26.61	31.94
Proportion of junior doctors in S1 (second row) that started medical school in 2006	0.76	0.68	2.21	23.85	20.83
Proportion of junior doctors in S1 (second row) that started medical school in 2007	0.00	0.34	1.33	0.00	22.22
Proportion of junior doctors in S1 (second row) that started medical school in 2008	0.00	0.00	0.00	0.92	3.39
Proportion of junior doctors in S1 (second row) that started medical school in 2009	0.00	0.00	0.00	2.75	3.57

- Line 130: Consider removing “(in a sort of pyramidal spread)”. I found that this clause confusing. Removed.

- Consider replacing with more fulsome description on how the Leslie matrices were created. Note, if this section starts to get too wordy, then consider creating a supplement with the full description. My understanding is that the table of students by education location (in which distance groups are nested) were used to derive the probability of movement to the next education location— this would be equivalent to the survival probability (S) of a typical Leslie matrix (see example below). If all cohorts were combined then the fertility probability (F) of a typical Leslie matrix would not be applicable and these cells in the matrix would be zero. If cohorts were treated separately, then my guess is that the fertility probability would be applied to the first stage only (i.e., incoming medical students). A

simplified Leslie matrix is shown below for the 3 stages—all cohorts combined. If I understand correctly, then the 25 km distance groups would be nested within the non-zero cells (specifically S0 and S1). Please confirm if this understanding is correct and add the salient details to the Methods (or Supplement).

$$\begin{bmatrix} & F1 & F2 & F3 \\ S0 & & 0 & 0 \\ 0 & S1 & & 0 \end{bmatrix}$$

We acknowledge the reviewer's observation that our initial explanation of the Leslie matrix lacked sufficient detail. In response to this feedback, we have incorporated a supplementary file [Supplementary information 3] outlining the step-by-step construction of the matrix, as suggested by the reviewer. Additionally, to alleviate potential confusion regarding the consideration of time, we have renamed the matrix to 'simplified stage-only Leslie matrix'.

- Line 130: Given that the matrix comprises counts of physician-distances, it is more accurate (albeit more wordy) to say “from home to medical school, from medical school to foundational training location, from foundation to specialization training location”. Questions: Can learners change locations during medical school, foundational training or specialty training? If yes, then how common is this changes? If yes and if this is common, then how were these changes captured in the data?

We have implemented the suggestion from the reviewer to name the distances from home to the medical school, from the medical school to the foundational training location, and from foundation to specialisation training location in both the abstract [Lines 28-30] and the results (Table 2). For the distances between the family home and the medical school, we have considered only the student's first choice. Out of 97,932 students, 5,093 (5.2%) changed medical schools. For the full pathway, we only considered those for whom the full pathway was available without changes. 45 junior doctors were excluded, and these include those who, at some point, were not in training or changed location during the foundation and specialty training. We have provided this information in the Data and Methods section: The distances between the family home and the medical school considered only the medical school student's first choice, and therefore, subsequent changes are excluded. Out of 97,932 students, 5,093 (5.2%) changed medical schools. The 871 individuals include only those for whom the full training pathway was available without changes. 45 additional junior doctors who, at some point, were not in training or changed location during the foundation and specialty training were excluded. [Lines 108-113]

and included as limitation in the Discussion: Other limitations are also present. This study includes information up to 2015, which may not reflect recent changes in medical education or workforce dynamics. Students who left their medical education or moved between training locations are not included in the dataset; therefore, the results are not generalisable for this group of students. [Lines 326-329]

- Line 141: Please clarify the data that were used to calculate the Gini coefficient and please explain how the line of equality was derived. Was the coefficient and line derived for all movement stages or just for home to medical school? The caption and axes titles for Figure 1 may need additional information on how the Lorenz curve and line of equality were derived.

We acknowledge that the previous description of the Gini coefficient and Lorenz curve lacked detail, and the legends of Figure 1 had incorrect axes labels. To address these issues, we have replaced Figure 1 with a new one (now Supplementary Information 6). The Gini coefficient was specifically applied to individuals who completed the full training pathway (871 students), and this clarification has been included in the Methods section: To confirm equality or inequality in medical students and junior doctors (871 individuals) movement distance categories, the Gini coefficient has been calculated which has been used in similar researches (11). [Lines 148-150]

Extended explanation of the Gini coefficient calculations has been provided: The Gini coefficient

varies between 0 and 1 where 0 implies complete equality, and 1 complete inequality. The coefficient is derived from the Lorenz curve. When the Gini coefficient is 0, the Lorenz curve follows the line of equality, which is a straight diagonal line. If the distribution exhibits any inequality, the Lorenz curve falls below the line of equality. The Gini coefficient is the ratio between the Lorenz curve and the equality line. The Lorenz curve is the line passing through the points of cumulative percentage of the population (on the x-axis) and cumulative percentage of distances (on the y-axis). [Lines 150-155]

- Given that schools and training locations tend to be located in urban areas and given that the population is predominantly located in urban areas, then presumably the line of equality should match the distribution of distances from the general population to medical school. In contrast, distances from medical school to foundation location would reflect a different distribution: distances between medical schools and foundation training locations weighted by the number of learners at medical schools and the number of positions available at the foundation training location. A similar argument would apply to the expected distribution of distances between foundation and specialization training locations.

The line of equality represents an equal distribution of all distance categories. In theory, if we have 10 categories, each should contribute to 10%. Therefore, the line of equality does not represent the 'urban distribution' of medical training organisations and the general population. The Lorenz curve may reflect the general population only if medical schools predominantly attract people living in urban areas. With this in mind, understanding how the distances between medical schools, foundation and specialty training deviate from individual movements is crucial. We have acknowledged this as a limitation of the study: The main limitation of our study is the absence of historical mapping for all distances between medical schools, foundation, and specialty training facilities. Currently, this information is unavailable, but it is crucial to test alternative hypotheses regarding the likelihood of movement distance in relation to the availability of training facilities in the surrounding area, weighted by the number of positions available at the training location. [Lines 321-325]

- Line 144, presumably the logistic Poisson GLM was applied to the "count of physician-distances" and not to the "distances" per se. Was this model applied to each mobility's stage or just to the home to medical school mobility stage? Was any consideration given to the hierarchical nature of some of the data (Specifically, individual level data, county level data)? A generalized linear mixed model might be a more appropriate statistical technique for dealing with hierarchical data. From the Methods it seems that a second Poisson analysis was applied to aggregated data. Please clarify.

The individual logistic Poisson GLM was applied to each student's 'family home to medical school distance', not to counts of physicians, because we aimed to estimate the relationship between distance and students' characteristics. The reviewer is correct in noting that in the county-level model (the second Poisson GLM), the outcome is the number of students moving between counties. In practice, we employed two distinct Poisson GLMs rather than a hierarchical model, as the explanatory factors (or covariates) differ between individual-level and county-level data. We have made corrections to the methods section to enhance clarity: Finally, a Poisson generalised linear model was applied individually on the distance (outcome) covered by each individual student from family home to medical school (therefore on the full 97,932 students). Distance was assumed a count data with discrete and non-negative values. The model explanatory factors were gender, ethnicity, and various variables representing the socio-economic status of the students and their parents (full list presented in Table 1). In addition, in the second Poisson generalised linear model, the number of students moving from the county of family home to the county of medical school was used as outcome. Since the data is aggregated by county, the model included the distance between training and family home counties and the modal values of each explanatory variable for both the departing (family home) and arriving (training) county. [Lines 156-164]

- Trivial, but if the Poisson analysis was the final analysis, then why is ANOVA mentioned as a separate analysis (here and in the Results)? Are the authors referring to the ANOVA component of the Poisson GLM? And that they used backwards-forward selection on the Poisson model? Please clarify.

The section on variable selection has been re-written: Before running the individual and county-level multivariable Poisson generalised linear models, selection of the explanatory variables was carried out by univariate modelling and stepwise selection (19) since all the explanatory variables are categorical. The variable selection approach included the following steps:

- 1) fit univariate models with each potential explanatory variable (Table 1);
- 2) on each univariate model a log-likelihood ratio test (comparing with the null model) is performed using the function `poisson.anova` in R-cran software(20) ;
- 3) the explanatory variable of the univariate model was included in the multivariable model if it caused the univariate model to be significantly different (at a significant level of 0.05) from the null model;
- 4) the multivariable model was then inputted in a stepwise selection method (function `stepAIC` in R-cran software(21)), with both forward and backward selection; the returned model with the lowest Akaike Information Criteria (AIC) (22, 23) was selected. [Lines 165-177]

- Results

- The results section is generally well-written but could use minor revision to improve clarity. Line 159: As noted previously, the subset of 871 individuals needs to be explained in the methods section and removed from the Results. For example, in the Methods write something like "Of 97,932 students, only 871 have data on the full path from family home to specialization."

This sentence has been moved to the methods and the numbers fully explained as described in the reply to the previous reviewer's comment: The data was provided from the UK Medical Education Database (UKMED) (project number UKMEDP44) and included 97,932 individuals who started medical school between 2002 and 2015. Of these, 871 had completed medical school and foundation training, and started or completed specialty training by the end of the study period (Supplementary File: Supplementary Information 1 and 2). We use the terms 'students' or 'medical students' to refer to those attending a medical school, and 'junior doctors' for those in foundation or specialty training. Since the dataset is only available up to 2015, the individuals for whom we have information for all three training stages are those who commenced medical school between 2002 and 2009. For all 97,932 students, the distances from the family home to medical school are available. [Lines 99-107]

- Lines 159-136: Remove these lines and move salient information into the Methods section. Line 164 becomes the first line in the Results section and it can be modified slightly to read "The analyses of the Leslie matrix, applied only to the 871 students for which the full pathway is known, found that..."

Sentences modified and moved to the methods (see reply to the last comment).

- Consider promoting supplemental tables A2 and A3 to become full table in the manuscript.

Figure 1 can become a supplement.

Figure 1 moved in supplementary (Supplementary Information 6). We have moved Supplementary Tables A2 and A3 in the main text (now Table 3 and Table 4).

- Lines 195-196: Sentence is incomplete...please revise.

Sentence removed because repeating information from the methods.

- Line 197: Is the model fitting "distances" or "counts of physician-distances", please clarify here and elsewhere.

Sentence removed; this is now explained in the methods (see reply to comment above).

- Lines 198-199: I strongly suggest using the brief variable name (a short descriptive name) rather

than the abbreviation. For example, “occupation category” rather than NSSEC, and “deprivation quintile” rather than IMD. Although the full explanation of the different variables and their categories belongs in Table 1, it would help the reader’s if a short descriptive name was used rather than referring the reader back and forth to Table 1.

Thank you for this suggestion. We have replaced NSSEC with student’s parents’ occupation and IMD with deprivation through the main text. See also Table 1 for revised variables’ names.

- Line 204: Please keep using the phrase “shorter distance” rather than “lower distance”. Please check and ensure consistency when referring to “distances”, “movements”, etc. As noted previously, technically the data are counts of students/physicians who have travelled specific distances (i.e., counts of physician-distances) and not distances per se. You can interpret the “count of physician-distances” as “distances” in the Discussion (with the appropriate caveat), but in the Results please use the precise term.

Lower replaced with shorter (sorry for this mistake). As explained above, we used distances for the first GLM and counts for the second one. Please refer to the reply to a similar comment above. We thoroughly examined the use of distances and movements throughout the manuscript and ensured consistency.

- Discussion

- The discussion is generally well-written with the salient material presented. Line 229: trivial, ascertain should be ascertained.

Corrected.

- Lines 230-231: Sentence is somewhat awkward and could do with revision (and a spelling check). Consider something like “The finding that the majority of UK-born [spelling] physicians in training travelled less than 50 km during 2002-2006 and in 2008 is reported here for the first time in the literature.”

Thank you, sentence changed as The up-to 50 km modal distance travelled by UK-born medical students and junior doctors for all the years between 2002 and 2009 is reported here for the first time. [Lines 269-270]

- Lines 231-233: Further clarification of what is reported in reference 23 would help. Did reference 23 speak specifically to foundational and/or specialty training location or was practice location included?

We have specified that these distances are between the medical school and their current job location: A 2019 GMC report that focused on doctors who gained their UK Primary Medical Qualification between 2012 and 2018 found that almost a quarter (23%) of doctors in 2019 were working within 10 miles (16 km) of where they studied at medical school (28). However, a third of graduates moved more than a hundred miles. [Lines 270-273]

- Line 249: consider replacing “lower number of doctors moving” with “fewer doctors moving”. And add more” to “more students with higher degrees at entry...”

Corrected.

- Lines 251-254: In summarizing findings from reference 26, is it correct to say that the students were “working” in deprived areas? Perhaps “studying” or “training” would be a better word choice. Are physicians still called “students” during foundational or specialty training? I ask because typically, the medical learner is a “doctor” and no longer a student once they finish their MD, MBBS, or equivalent.

Regarding the word ‘working’ we are using the same terminology from the cited publication which states: GPs whose parents had semi-routine or routine occupations had 4.3 times the odds of working in a deprived practice compared to those with parents from managerial and professional occupations.

The point regarding ‘students’ versus ‘doctors’ is indeed important. For simplicity, we use the term ‘students’ to refer to all those attending a medical school, while ‘junior doctors’ pertains to individuals in foundation and specialty training. We have clarified this in the data and methods section: We use the

terms 'students' or 'medical students' to refer to those attending a medical school, and 'junior doctors' for those in foundation or specialty training. [Lines 102-104]

- Line 259: Consider revising this sentence as “that the geographic distribution of ethnic minorities in UK is clustered (27)”

Sentence changed in: although our analyses do not account for the potential geographical clustering of ethnic minorities in the UK (32) which can potentially bias the results. [Lines 299-300]

- Lines 267-269: Please add more detail to clarify how access to medical care, childhood background and the class of population that the GP is likely to serve is connected. As noted earlier, if physicians stay close to home or stay close to where they trained, then the maldistribution of physicians in the UK will reflect the applicant pool/admissions process, and the location of medical schools or postgraduate training sites rather than the distribution and care needs of the population. There is considerable evidence from the UK and internationally supporting the selection of students from underserved areas and the location of training sites in underserved areas as one approach to addressing this maldistribution. See, for example WHO 2021 <https://apps.who.int/iris/handle/10665/341139> I would also mention that financial incentives have had mostly limited and short-term success in recruiting physicians (or healthcare workers in general) to underserved areas and retention tends to be poor.

We have added the following paragraph in the Discussions: In Scotland it has been found that there is a link between childhood background, as defined by parental socioeconomic status and being in more remote residence, and the population the GPs are likely to serve(31). The authors suggested that GPs residing in more rural areas and those who experienced greater deprivation during their childhood tended to work in settings that reflected these characteristics. While this should not be presumed from these students, World Health Organisation recommendations emphasise the need to create more opportunities, such as locating educational facilities closer to rural areas, and providing incentives, such as benefits or direct payments, to medical students and doctors to serve in rural areas (37). However, recent literature indicates that outcomes of financial incentives are limited or unclear (38). Family-unit and community considerations are often the predominant factors influencing doctors' movements (39). [Lines 310-319]

- Line 270-271: This sentence is awkward and requires revision. Consider “the present study did not consider international medical graduates who work in the UK, or UK-born [spelling] doctors working abroad: data for these physicians were not available.”

Sentence changed: Finally, the present study did not consider international medical graduates working in the UK, UK-born doctors working abroad, or records pre-2002. Data for these doctors were not available. [Lines 331-332]

- Line 274: “individual beliefs”[plural] and “The proposed study...” rather than “This study...”. Corrected.

- Line 285: Acknowledgements. Should “I am grateful...” be revised to read “We are grateful...” ? Corrected.

- References

- The range of references seems appropriate to the topics covered in the manuscript. However, there are a number of references with incomplete information or have incorrect formatting: journal titles are missing (e.g., #8, 14, 15); different abbreviations or different names are used for journals; citation information is missing (e.g., #28) and capitalization is inconsistently or incorrectly applied to proper nouns (e.g. #8 “british doctors”) and to journal names/abbreviations. Citations for IBM SPSS and STROBE should be added.

Corrected.

- Tables and Figures

- Table 1: Additional information is needed. It would be best not use variable names in the first column given that variable names only have meaning for the research team. Please use brief

descriptive names in the first column (Variable) and ensure that the Description has the full name (not the variable name) and abbreviation. The four distance measures can be combined as indicated below. Similarly for Ethnicity, Income, Occupation (NSSEC?) and please group similar variables together (e.g., Ethnicity with BME, Occupations-NSSEC).

Variable	Description
Distances	Road distances (? straight-line distances ?) from: 0-home county to medical school county 1-home to medical school 2-medical school to foundation training location 3-foundation training location to specialty training location.

A new table 1, with full explanations, is provide and acronyms are removed:

Table 1: List of variables used in the analyses and their definitions. Source: UKMED data dictionary(25).

Variable	Description
Individual Student-related variables	
Bursary	Whether the individual is in the UKCAT Bursary table
Distance	Coded as the travel distance between centroids: 0 = from the county of family home to the county of the medical school; 1 = from home postcode to the medical school postcode; 2 = from medical school postcode to foundation training postcode; 3 = from foundation training postcode to specialty training postcode.
Domicile County	The county the student family home is.
Ethnicity	Level 1 and Level 2 classification of ethnic groups as in 2011 Census. Full list in Supplementary File: Supplementary Information 4.
BME	Black and Minority Ethnic (BME) (categorised 1 and white 0)

Gender	The gender of the student categorised as: 0 male and 1 female
Medical School County	The county the medical school is in.
Qualification	The highest qualification the student has on entry to medical school: 01 = no formal qualification/not known 02 = level 2 and below 03 = level 3 including A levels and highers
School Type	Type of school the student attended between the ages of 11 and 16. Classes are: 01 = independent/Private Fee Paying School; 02 = grammar School (selective); 03 = academy/Secondary School; 04 = further Education College / Sixth Form College; 05 = Other.
Socio-Economic Classification	Socio-economic background of students aged 21 and over at the start of their course, or for students under 21 the socio-economic background of their parent, step-parent or guardian who earns the most is returned. It is based on occupation, and if the parent or guardian is retired or unemployed, this is based on their most recent occupation. Classes are: 01 = higher managerial & professional occupations 02 = lower managerial & professional occupations 03 = intermediate occupation 04 = small employers & own account workers 05 = lower supervisory & technical occupations 06 = semi-routine occupations

	07 = routine occupations 08 = never worked & long-term unemployed 09 = not classified
Individual student's parents-related variables	
Deprivation	Index of multiple deprivation (IMD) of the zone the individual is from where 1 = most deprived and 5 = least deprived The IMD is the official measure of relative deprivation in England. It is based on 39 separate indicators, organised across seven domains of deprivation (income; employment, health deprivation and disability; education, skills training; crime; barriers to housing and services; and living environment. The full rank is divided in five quintiles, with quintile 1 containing the most deprived or poor and quintile 5 containing the least deprived or most affluent.
Income Support	Whether the student's home has had any income support during their time at school.
Occupation	Occupation of the household reference person. This is the person responsible for owning or renting or who is otherwise responsible for the accommodation. In the case of joint householders, the person with the highest income takes precedence. Where incomes are equal, the oldest person is taken as household reference. Based on the National Statistics Socio-Economic Classification: 1 = managerial and professional occupations 2 = intermediate occupations 3 = small employers and account workers 4 = lower supervisory and technical occupations 5 = semi-routine and routine occupations
Occupation individual	Socioeconomic class occupation description of parent 1 and parent 2 from The UK Clinical Aptitude Test (UKCAT) questionnaire. Classes are:

	01 = modern Professional (e.g. teacher, nurse, social worker, artist, police officer (sergeant or above)); 02 = clerical & intermediate (e.g. secretary, call centre agent, nursing auxiliary, nursery nurse); 03 = senior managers or administrators (e.g. finance manager, chief executive); 04 = technical & craft (e.g. motor mechanic, plumber, printer, tool maker, gardener, train driver, fitter); 05 = semi-routine manual & service (e.g. postal / farm worker, security guard, catering/sales assistant); 06 = routine manual & service (e.g. HGV driver, cleaner, porter, sewing machinist, bar staff, labourer); 07 = middle or junior managers (e.g. office / retail / bank / restaurant / warehouse manager, publican); 08 = traditional professional (e.g. accountant, solicitor, medical practitioner, scientist, civil servant); 09 = never worked; 10 = don't know or prefer not to say.
--	---

- Distance: is this road distance based on exact addresses, based on postal code centroids, or straight-line Euclidean distance to specific locations? or to centroids?

We have amended table 1 to explain that origin and destination points are referred to the centroids of the postcodes. The distance is the travel distance (see Table 1 above).

- Please define HESA and SEC.

Full name provided.

- NSSEC when first mentioned applies to whom, the student? The student's parents? Please clarify.

Now NSSEC has be labelled 'Occupation' and refers to the household reference person as described in Table 1.

- "Quintile" needs additional information in the name. Consider labelling this variable "Deprivation".

We have renamed the variable Deprivation.

- SEC: how does this variable differentiate between the student's classification when the student is over 21 or the student's parents classification (when the student is under 21 years of age)?

It is referred to the parent/guardian as described in the revised Table 1.

- Occupation 1 and 2: presumably this is the NSSEC value for the parents. Please clarify.

Renamed 'Occupation individual' and provided for parent 1 and parent 2. Full description in Table 1.

- School type: please list the type of school categories.

School categories provided (Table 1).

- Please provide a reference for these variables. If variables are fully defined in the UKMED data dictionary, then it should suffice to reference the data dictionary.

Reference provided (number 25):

https://www.ukmed.ac.uk/documents/archived_data_dictionary/UKMED_Data_Dictionary_02_2018.pdf

- Figure 1: The caption and axes titles may need additional information on how the Lorenz curve and line of equality were derived. Consider moving this figure to the supplements.

The figure has been modified as described above (now Supplementary Information 6):

- Figure 2: Is the table supposed to be asymmetrical? I ask because if you read the table from top to bottom, then you do not get the results that the authors describe. If the upper left diagonal was deliberately removed, then I suggest using white (blank) cells and explaining this feature in the figure caption.

No, the table is symmetrical. Sorry about the confusion with the colours. We have modified it:

- Figure A3: Trivial, but please replace (up) and (down) with (top) and (bottom) or label as (a) and (b). Both graphs are “up” from the caption. Please correct the spacing in the caption.

Corrected.

- Tables A2 & A3: Consider moving these tables to the body of the manuscript. Please use meaningful labels rather than the variable names. Otherwise you force the reader to go back and forth between Table 1 and the Results tables. “Estimate” should be relabeled as “Coefficient” I assume that this the raw coefficient (i.e., not standardized). Please clarify. Consider adding standardized coefficients to help the reader assess the relative importance of each variable. Formatting of values in the table should reflect common usage. While I appreciate the use of scientific notation and the 4 significant digits, most readers would benefit from seeing the values in straight forward decimal format with 3 digits (e.g., -0.963, -0.006). Formatting of p-values should reflect common usage: $p < 0.001$ (< 0.0001) for the smallest values should suffice.

Table A2 and A3, now referred to as Table 2 and Table 3 in the main text, have been updated: the full names of predictors have been provided, numerical values corrected to three digits, and p-values adjusted to reflect common usage. Coefficients have not been standardized, as the majority of the predictors are categorical.

Reviewer: 2

Dr. Sarah Holdren, The University of North Carolina at Chapel Hill School of Medicine

Comments to the Author:

Thank you for allowing me to review this interesting work. The description of the study is excellent and fills a need in the literature. There are only a few minor typos that will need edited during copy editing.

We appreciate the reviewer for their positive assessment and for emphasizing the significance of our work. We have revised the manuscript, checking for typos or unclear sentences.

Reviewer: 3

Dr. Leslie Foldager, Aarhus Universitet, Aarhus Universitet

- It appears to me as an interesting and important investigation. I think parts of the manuscript could be improved a bit and at least some of the analyses are not described clearly enough, leaving doubts as to what was actually done. I am sure this can be handled without re-doing everything though some consideration may be needed regarding the relatively many categories of distance. Would it be possible to include more students by omitting some variables? or is it the information on the actual movement (and distance), which is missing? I really miss some explanation why it was impossible to get individual data, at all, from the last six years (2010-15).

We thank the reviewer for these important points. By removing variables, we would not increase the number of students for which we know the full training pathway. This depends only on the time necessary for a medical student to obtain a degree and commence foundation training. Foundation training is two years, after this the student will commence specialty. Therefore, the students registering in 2009 are the last ones for which we know the full pathway (from medical school to starting specialty). We have added this information in the methods: The data was provided from the UK Medical Education Database (UKMED) (project number UKMEDP44) and included 97,932 individuals who started medical school between 2002 and 2015. Of these, 871 had completed medical school and foundation training, and started or completed specialty training by the end of the study period (Supplementary File: Supplementary Information 1 and 2). We use the terms 'students' or 'medical students' to refer to those attending a medical school, and 'junior doctors' for those in foundation or specialty training. Since the dataset is only available up to 2015, the individuals for whom we have information for all three training stages are those who commenced medical school

between 2002 and 2009. For all 97,932 students, the distances from the family home to medical school are available. [Lines 99-107]

- This study aimed to understand movements (or distance of movements) and spatial patterns of medical student throughout their training. The study found that the students do not move far but with an inhomogeneous pattern over time and within individual characteristics. Statistical analyses were integrating different methods and applied on both individual and aggregated movements. Six methods have been used for various parts of the analyses: 1) Leslie model (Leslie matrix), 2) PCA, 3) Gini coefficient, 4) chi-squared test, 5) GLM, and 6) ANOVA. The Leslie matrix was analysed by use of PCA and comparing resulting eigenvectors through chi-squared test. Moreover, Gini coefficients were calculated to confirm presence of dominant distance categories. The GLM concerned (at least) the distances travelled by the individual students. Was it using the distance as such as response? (not clear to me from L144-145)

Yes this is true for the first Poisson GLM. The text has been changed to clarify this: Finally, a Poisson generalised linear model was applied individually on the distance (outcome) covered by each individual student from family home to medical school (therefore on the full 97,932 students). Distance was assumed a count data with discrete and non-negative values. The model explanatory factors were gender, ethnicity, and various variables representing the socio-economic status of the students and their parents (full list presented in Table 1). In addition, in the second Poisson generalised linear model, the number of students moving from the county of family home to the county of medical school was used as outcome. Since the data is aggregated by county, the model included the distance between training and family home counties and the modal values of each explanatory variable for both the departing (family home) and arriving (training) county. [Lines 156-164]

- The study was based on registry data from the UK Medical Education Database (UKMED), including 97,932 students who started medical school (one of 36 schools) between 2002 and 2015. Distances were available for 3 stages on the individual level: 1) family home to medical school, 2) medical school to foundation, and 3) foundation to specialisation. In addition, distance from the county of the family home to the county of the medical school was available, and two variables identifying these counties. Moreover, several variables related to gender, ethnicity, socio-economy, etc. (listed in Table 1). Re-reading my comments I wonder if only distances from family home to medical school was analysed? or was this just in relation to the county-based analysis?

We sincerely apologize for any confusion caused by the lack of clarity in the previous version, which was also noted by reviewer 1. The initial set of methods was applied to the entire pathway, while the Generalized Linear Models (GLMs) specifically targeted distances from family homes to medical schools. We have addressed and clarified this point: Six methods were used to investigate the movements of medical students across the UK. Four of these methods were integrated to investigate the movement distances and their patterns across the three stages and years: the Leslie model (known as a Leslie matrix), principal component analysis (PCA), Gini coefficient and chi-square test. Generalized linear models (GLMs) and variable selection methods were employed to evaluate the factors influencing the movements from family home to medical school (which provide the largest data) at both individual and geographically aggregated levels. In practice, the first four analyses (Leslie, Gini, PCA and chi-square test) were applied to the 871 records containing the full pathways; while the Poisson GLMs and variable selection methods were applied on the whole 97,932 movements from family home to medical school. [Lines 122-130]

- Table 1: I do not think it is totally clear in all cases whether the variable concerns the student or the student's parents/family. The variable 'Gender' would be more clearly denoted 'Female'.

Table 1 has been improved and we have separated variables related to students from the variables related to the student's parents. We kept 'Gender' as usually this variable name is applied in similar analyses. See Table 1 above in response to reviewer 1.

- L127-128: Just wondering that you say the method was first developed in the 1960s but the reference (no. 15) is from 1945!? The reference is also not complete in the ref. list ... it is from *Biometrika*. Generally, the reference list is rather messy and incomplete!

We apologise for the issues we had with endnote. We have rectified the reference list.

- L144: What is a 'logistic Poisson' GLM? The Poisson family does to my knowledge allow for a log, identity or sqrt as link function. Did you run a quasi-Poisson model? Moreover, you say 'distances' in plural ... so did each student contribute with more than one observation to this model? In that case how about correlations among repeated observations from the same student?

Indeed, 'logistic' is not correct, and it should have been referred to as the 'log link function' instead. We have removed 'logistic'; that was the only instance where 'logistic' was mentioned, and it was a typo. Since each student contributed to only one distance (from family home to medical school), we are not dealing with within-student correlation. We have also corrected the term to 'distance' in singular form. Here is the updated version: Finally, a Poisson generalised linear model was applied individually on the distance (outcome) covered by each individual student from family home to medical school (therefore on the full 97,932 students). Distance was assumed a count data with discrete and non-negative values. The model explanatory factors were gender, ethnicity, and various variables representing the socio-economic status of the students and their parents (full list presented in Table 1). In addition, in the second Poisson generalised linear model, the number of students moving from the county of family home to the county of medical school was used as outcome. Since the data is aggregated by county, the model included the distance between training and family home counties and the modal values of each explanatory variable for both the departing (family home) and arriving (training) county. [Lines 156-164]

- L147: To me it sounds like both this model and the model in L144 might be over-dispersed, and this may be a reason for choosing a quasi-Poisson model. But why a logistic link as opposed to say a negative binomial model with log link? I miss some more description of the considerations behind the model choices.

With the new edits we have clarified that we employed a Poisson GLM. The Poisson regression was chosen as providing the lowest AIC when compared to a negative binomial model (which account for overdispersion). We have added this information to the text: AIC was also employed to compare the Poisson regression model with a negative binomial model, which can account for overdispersion. The former yielded a lower AIC (indicating a better fit) than the negative binomial model, and for this reason, only the Poisson regression results are shown. [Lines 179-181]

- L150-151: It seems counterintuitive to talk about 'univariate modelling' in the context of backward-forward selection, which would normally be used to build multivariable models. Moreover, I would call it 'forward selection' and 'backward elimination' (as you are not selecting variables but rather deselecting, i.e., eliminating variables, in the backward part). The process is so well known that I do not think a reference is needed – however, you need to state your thresholds. Also, if you start from some model with several variables (which could be one way to go), then state what the initial model include. If you start with the null model (and adds to this in a forward manner, followed by backward elimination) than state this. How about interactions?

- L151-152: To me ANOVA means that a normal distribution of residuals has been assumed (i.e., a normal linear model) but you have just described Poisson models so this (univariate?) testing of explanatory variables by ANOVA is by no means clear to me. Or did you perhaps mean that the

ANOVA was used for the county-based analysis? This is however not what one will understand from 'the same model' in L147.

- L153: So, you did not test the increase in variance mentioned in L152 or did you just look at the AIC and included an extra variable if the AIC decreased? Or how is this to be interpreted? Moreover, here it sounds as if you choose among several models, but this is not as such the case if you used a "forward selection backward elimination" procedure ... rather this is a model building procedure. AIC can be used for this and sometimes is but then you are not lining the models up and choosing. AIC is also sometimes used for decision when models are not nested – but then it would not be a forward/backward process.

- L150-153: All together, I find the description unclear and I am not sure what you actually did.

The four comments are grouped since regarding the variable selection method. First, sorry for the shorthand in writing this section, we very briefly mentioned the steps but not enough to be fully understood. In the revision, we have detailed the variable selection method and what we meant for ANOVA: Before running the individual and county-level multivariable Poisson generalised linear models, selection of the explanatory variables was carried out by univariate modelling and stepwise selection (19) since all the explanatory variables are categorical. The variable selection approach included the following steps:

- 1) fit univariate models with each potential exploratory variable (Table 1);
- 2) on each univariate model a log-likelihood ratio test (comparing with the null model) is performed using the function `poisson.anova` in R-cran software(20) ;
- 3) the explanatory variable of the univariate model was included in the multivariable model if it caused the univariate model to be significantly different (at a significant level of 0.05) from the null model;
- 4) the multivariable model was then inputted in a stepwise selection method (function `stepAIC` in R-cran software(21)), with both forward and backward selection; the returned model with the lowest Akaike Information Criteria (AIC) (22, 23) was selected. [Lines 165-177]

- L159-161: How did you determine that the 871 students are representative for the full dataset? Did you just look at the histograms and then by eye decide for yourself that this looks fine? Wouldn't it be possible to test? For example, how about splitting the data into two (1) the 871 and (2) the rest, and then formally test if the distributions are not different (perhaps by non-parametric statistics if it does not make sense to do a parametric model)? Do you have any explanation why paths are only complete for (some) students from years 2002-2009? Why are the last six years less complete?

We decided to remove this comparison since the two datasets are used to answer different aims: the 871 students to identify patterns in student's movements during their pathway, and the 97,932 students to estimate important factors associated to the movements between family home and medical school. As described previously, the information regarding why for some students we have the full pathway and for others not as been provided: The data was provided from the UK Medical Education Database (UKMED) (project number UKMEDP44) and included 97,932 individuals who started medical school between 2002 and 2015. Of these, 871 had completed medical school and foundation training, and started or completed specialty training by the end of the study period (Supplementary File: Supplementary Information 1 and 2). We use the terms 'students' or 'medical students' to refer to those attending a medical school, and 'junior doctors' for those in foundation or specialty training. Since the dataset is only available up to 2015, the individuals for whom we have information for all three training stages are those who commenced medical school between 2002 and 2009. For all 97,932 students, the distances from the family home to medical school are available. [Lines 99-107]

- L167: If 54% of the 871 students happens to be in the 0-25 and 25-50 km categories, then there must be some of the other 38 categories that include very few (perhaps even zero) students!?

Splitting this on years will obviously give even fewer in each category. How robust are these analyses to zero cells or cells with very few expected counts? Do the categories have to be with same lag (25 km) or would collapsing categories be a way to avoid having (too?) few students in some categories?

While we appreciate the reviewer's concern, it is essential to note that if the goal was to achieve similar sample sizes, aggregating data across all years would inevitably lead to dissimilar aggregations, resulting in incomparability and rendering the interpretation of general results impossible. The current grouping strategy was chosen deliberately as it enabled the identification of spatial scales (100 to 150 km) that might have been obscured or lost with a different aggregation approach. PCA is sensible to zero-inflated variables, however most of the zeros are homogeneously distributed across all cohorts to the largest distances for each cohort. This homogeneity reduces the sensitivity of PCA to the presence of zeros.

- L170: Here and a couple of other places you use the words 'equity' and 'inequity'. I guess (most of) you have English as your native language, so I almost do not dare questioning this. Nevertheless, did you want this to be 'equity' and not "equality"? In Fig. 1 you use 'equality' and 'inequality' so at least you leave some unclarity for us non-native English speakers.

Yes, it is equality and inequality. We have corrected these. The issue was that a publication (that we cited several times) was using inequity, when it should have been inequality.

- L182-186 and Fig. 2: Do you not have a multiple comparison problem here that should be accounted for? Perhaps you should mention (to make this clear to all) that those in red are interpreted as having significantly different proportion(s?) of short movements.

The significant difference in proportions cannot be attributed to only short movements (Supplementary File 5). We improved clarity in the meaning of the red cells of the matrix: Figure 1 shows that 2002 proportion of movements are significantly different from most of the rest of the years (in red) at a significant level of 0.05, apart from 2007. [Lines 211-212]

- Supposedly the matrix should be symmetric and thus, if presented, the upper left triangle should have the same colours as the lower right triangle. Wouldn't it be better to leave out one of the triangles? Moreover, I suppose comparing a year with itself is not meaningful so the diagonal should also be left empty. In the figure text you could mention what you understand by 'first components'.

Figure 1 (previously Figure 2) has been modified to account for the reviewers' comment and to improve clarity (see attached figure 1 in reply to reviewer 1).

- Likewise, it is not clear what you understand by 'short movements' (L182). Is it only the 0-25 category? or is it both 0-25 and 25-50? or ...?

'Short movements' has been replaced with 'movements' because while the 0-25 is the dominant category for most of the years, it is not the only one contributing to the first principal component extracted from PCA. To clarify how Figure 1 has been generated we have added the following text: In other words, annual Leslie matrices were created, and the relative first principal components (containing the largest amount of explained variance) for each year were extracted and compared in pairs by chi-square test. [Lines 145-147]

- Table A2: I would think that the overall test (p-value) for the variable would make more sense to present and then by letters indicate which levels are different/not different when compared pairwise ... after relevant adjustment for multiple comparison, of course. Regarding small values, either you have to superscript '-16' (and other equivalent values) or use the E-notation, e.g., $<2e-16$ (personally I prefer having a white space between '<' and the number, i.e., $< 2e-16$, but this is just a preference). Another personal preference is to present standard errors with 1 more decimal than the means. 'Quintile' ??? quintile of what?

Table A2 (now referred to as Table 3 in the main text) has been updated. As the variables are categorical, providing an overall test p-value is not feasible due to singularities. Following the recommendations of both reviewers, we have adopted a more common approach in presenting the p-values and have modified the labels of the predictors:

Table 3: Poisson generalised linear model results for individual movements. Occ: occupation of the household reference; REF: reference category.

Variable	Coefficient	Standard Error	p-value
Intercept	5.197	0.015	< 0.001
Occ. managerial	REF		
Occ. intermediate	-0.086	0.002	< 0.001
Occ. small employers	-0.078	0.002	< 0.001
Occ. lower supervisory	-0.111	0.003	< 0.001
Occ. semi-routine	-0.121	0.003	< 0.001
Deprivation (Level 1)	REF		
Deprivation (Level 2)	0.011	0.001	< 0.001
Deprivation (Level 3)	-0.036	0.001	< 0.001
Deprivation (Level 4)	-0.069	0.001	< 0.001
Deprivation (Level 5)	-0.211	0.002	< 0.001
African	0.036	0.015	0.016
Arab	-0.244	0.015	< 0.001
Bangladeshi	-0.262	0.015	< 0.001
British	0.101	0.015	< 0.001
Caribbean	-0.016	0.016	0.325
Chinese	0.063	0.015	< 0.001
Gypsy or Irish Traveler	0.180	0.042	< 0.001
Indian	-0.157	0.015	< 0.001
Irish	0.347	0.015	< 0.001
Pakistani	-0.222	0.015	< 0.001
White and Asian	0.075	0.015	< 0.001
White and Black African	0.134	0.016	< 0.001
White and Black Caribbean	0.092	0.016	< 0.001
Other Asian background	-0.197	0.015	< 0.001
Other Black background	0.035	0.021	0.089
Other Mixed background	-0.046	0.015	0.002
Other White background	0.108	0.015	< 0.001
Other Ethnic group	-0.170	0.015	< 0.001
Not stated	0.081	0.022	< 0.001

- L204 and L206: It is not clear what you mean by 'lower distances' and 'longer distances' ... where is the cut?

Sentence rephrased: In terms of the ethnicity variable, it was possible to identify that Arab, Bangladeshi, Indian, Pakistani, 'other Asian background', 'other mixed background' and 'other ethnic group' students travelled shorter distances to reach the medical school than British, Chinese, Gypsy or Irish traveller, Irish, Scottish, White and Asian, White and Black Caribbean, White and black African, and 'other white background' students. Within these two groups of ethnicities distinguished for their relationships with travelled distance, a three-fold difference was found between British and African in their longer movements (with British moving further), while a 1.6-fold difference was found between Arab and Indian in their shorter movements (with Arab staying closer). [Lines 236-243]

- Table A3: A number like -9.629×10^{-1} is better presented as -0.9629 (I would probably say that 3 decimals, i.e., -0.963, is sufficient) and correspondingly, 5.862×10^{-2} as 0.5862. You could perhaps write e.g., < 0.0001 instead of 3.175×10^{-5} . At least remove the prefix 0's and as mentioned for Table A2, use superscript. It is not super clear what is meant by 'Quintile Depart' and 'Quintile Arrive' ... is it a 0/1 dividing into $\leq / >$ the quintile of depart (arrive) distance? Perhaps add a footnote explaining this. 'Gender Depart' ??? What do you mean by 'statistical summary of the explanatory variables' coefficient'? Isn't it the estimate from a model? If not, what is it then? I do not think it is possible to understand this table stand-alone. Perhaps it is by consulting the methods section but a variable like 'Gender Depart' is mysterious to me ... how is this defined for the number of students leaving home from a specific county? Moreover, if you do not change 'Gender' to Female (see comment to Table 1), you will need to tell that Gender=1 are females and Gender=0 (the reference) are males. If any of the variables are continuous, we need to know the unit for the estimate/coefficient. If it is not continuous, what is then meant by e.g., the 'Distance' variable? and what are reference categories for those that are dichotomous or categorical?

Table A3 (now Table 4 in the main text) has been changed to account for reviewer comments:

Table 4: Poisson generalized liner model results for county-aggregated data. Departing = county of family home; arriving = county with the medical school.

Variable	Estimate	Standard Error	p-value
Intercept	-0.963	0.068	< 0.001
Distance between counties	-0.006	0.000	< 0.001
BME departing county (No)	REF		
BME departing county (Yes)	-1.204	0.042	< 0.001
Gender departing county (Male)	REF		
Gender departing county (Female)	2.691	0.029	< 0.001
Qualification departing county (below level 3/not known)	REF		
Qualification departing county (level 3)	0.085	0.004	< 0.001
Deprivation departing county (level 1)	REF		
Deprivation departing county (above level 1)	-0.454	0.005	< 0.001
Deprivation arriving county (level 1)	REF		
Deprivation arriving county (above level 1)	-0.483	0.006	< 0.001

Students socio-economic (higher managerial)	REF		
Students socio-economic (lower and below)	-0.183	0.021	< 0.001

- L285: 'I am grateful' ... I hope all authors are grateful – shouldn't this be 'We are grateful' or 'The authors are grateful'? This would also correspond better to 'for their analysis' ... I hope UKMED bears responsibility for their own analysis though if this was what you meant.

Thanks, corrected.

VERSION 2 – REVIEW

REVIEWER	Foldager, Leslie Aarhus Universitet, Dept. of Animal Science
REVIEW RETURNED	29-Jan-2024

GENERAL COMMENTS	I think the authors have done a good job of revising according to not least the excellent review that you got from reviewer #1. It is now much clearer what was done, how variables and data sets are “defined” and which limitations you faced. The answers to my inquiries and comments are almost satisfactory, see the specific comments below. A few specific comments:  • You need to decide if you want to write ‘generalized’ or ‘generalised’ in ‘generalised linear models’ and correct accordingly. Personally, I would use ‘generalised’ unless the journal specifically asks for American English. • L31: personal preference ... I would not use capital letters in ‘principal component analysis’ and I would use plural “components” (the latter part applies everywhere in the manuscript) • L117: ‘individual’ typo • L160: I think ‘the second’ should be “a second” • L171 and L175-6: ‘R-cran’ should just be “R” ... CRAN (The Comprehensive R Archive Network) is the archive from which various version of R and packages can be downloaded. Usually, one would also refer which version of R was used, e.g. “R v.4.3.0” or “R-4.3.0” • Reference 20: the correct reference to the R software is (e.g., version 4.3.0 ... the only thing changing between versions is the year): R Core Team (2023). R: A Language and Environment for Statistical Computing. R Foundation for Statistical Computing, Vienna, Austria. https://www.R-project.org/. • L179: Since the negative binomial model is a Poisson model with an extra parameter taking account for the (over-)dispersion, i.e., the two models are (in some sense) nested, it is also possible to test the reduction from negative binomial to Poisson by a likelihood
---

ratio test if the function used can do both (e.g., glmmTMB). Some functions/packages/software also present a test for the dispersion parameter. Nevertheless, you would probably get to the same conclusion.

- Table 1: 'family home is' ... should 'is' be followed by "in"? I suggest that the UKCAT is defined in the first row (Bursary) instead of in the 'Occupation individual'-row.

- L200: I do not think 'C' should be capital in 'Gini Coefficient'.

- Supplementary Table S2: 'Proportion' is somewhat misleading as you are showing percentages ... so either "Proportion (in %)" (or "Proportion (in percentage)") or add "%" to all numbers in the table (from 3rd row and down) or show proportions, i.e., 0.1439 instead of 14.39.

- Supplementary Figure S1: In the version I look at, the 'bold black line' is red not black ... but the colour is not need as you can just refer to the full line and the broken line (diagonal).

- Regarding your reply to my comment (reviewer #3) on Table A2 (now Table 3): If you can run the model on the categorical variable, for sure you could also get a p-value. Using R software, you would simply run the model with (m1) and without (m2) the specific categorical variable and use `anova(m1,m2)`. Nevertheless, now I see that for ethnicity (shouldn't there be a heading 'level 2 ethnicity' or mentioning in the figure text?) you do not have a REF category. Indeed, then the model is overparameterised and perhaps this will lead to a singularity error. It seems that you present all possible level 2 ethnicities plus the rest 'Not stated' so if you model included these 19 indicator variables, the two categorical (deprivation and Occ) and the intercept, then this is also overparameterised and should lead to the same problems of singularity ... well, unless there is a 'rest' group in addition to 'not stated', i.e., a 20th group. However, if you make the 19-leveled categorical variable and chose a REF category (perhaps British would be most obvious) then this should be the exact same model as presented and an overall test of this variable will be possible (using `anova(m1,m2)` as explained above). You did not give a reply to the concern of multiple testing.

- Supplementary information 4: level 1 ethnicity groups are not shown in bold font (as far as I can tell).

- Regarding the reply concerning collapsing distance categories (see e.g., Suppl. Table S3): even in the 'Overall' there are small proportions, and you could just decide the collapsing on basis of the Overall and then let it be a limitation if some cohorts would have small proportions for some categories. The "problem" would at least be smaller than with the current categorisation. I can see that you have an issue in 2009 for the 100-125 group but you could perhaps then keep this. For the longer distance I do not think there is any gain in the finer grid as the information behind is weaker due to lower counts. Nevertheless, this is not crucial for me.